# Off-shell strings I: S-matrix and action

**Amr Ahmadain[⋆] and Aron C. Wall[†]**

Department of Applied Mathematics and Theoretical Physics (DAMTP),
University of Cambridge, Cambridge, United Kingdom

⋆ amrahmadain@gmail.com , † aroncwall@gmail.com

## Abstract

We explain why Tseytlin's off-shell formulation of string theory is well-defined. Although quantizing strings on an off-shell background requires an arbitrary choice of Weyl frame, this choice is not physically significant since it can be absorbed into a field redefinition of the target space fields. The off-shell formalism is particularly subtle at tree-level, due to the treatment of the noncompact conformal Killing group $SL(2,\mathbb{C})$ of the sphere. We prove that Tseytlin's sphere prescriptions recover the standard tree-level Lorentzian S-matrix, and show how to extract the stringy $i\varepsilon$ prescription from the UV cutoff on the worldsheet. We also demonstrate that the correct tree-level equations of motion are obtained to all orders in perturbation theory in $g_s$ and $\alpha'$, and illuminate the close connection between the string action and the c-theorem.

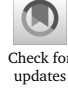 Check for updates

# 1   Introduction

It is widely believed by many in the string theory community that nobody knows how to make sense of off-shell string theory, and that a consistent definition may not even exist [1, 2]. But this is incorrect. As we hope to demonstrate in this article, there exists a viable formulation of off-shell string theory, which was pioneered most notably by Tseytlin in a series of papers [3–5] on the "nonlinear sigma model" approach to string theory.[1]

In general, an off-shell approach to string theory has multiple advantages: 1) it allows you to directly derive a target space effective action from the string worldsheet (rather than having to deduce the action indirectly from the equations of motion); 2) it allows you to discuss e.g. n-point correlators of massless fields, without having to take the LSZ limit where the insertions go off to infinity; 3) it allows you to invoke intrinsically off-shell constructions, most notably off-shell calculations of black hole entropy, which require introducing a conical singularity that violates Einstein's equations.

---

[1]Tseytlin's approach is distinct from string field theory, which has a different method of going off-shell. In particular Tseytlin's approach is able to define sphere diagrams with less than 3 insertions, which is a difficult problem in string field theory [6].

In the present paper (part I), we will give an accessible overview of Tseytlin's off-shell formalism, and will provide a general proof that it gives the correct tree-level S-matrix and equations of motion, to all orders in $g_s$ and $\alpha'$. In part II of this work [7], we will explain how this formalism was used by Susskind and Uglum (S&U) to calculate black hole entropy [8].

Although we are deeply indebted to previous research on this subject, this work is *not* intended as a review article highlighting past accomplishments. Instead our goal is to explain the off-shell formalism in our own language, to significantly generalize its scope, and to explain more carefully its conceptual justification. In particular, we improve on Tseytlin's work by using conformal perturbation theory to perturb around arbitrary (possibly strongly coupled or highly non-geometrical) string backgrounds. Our proofs that the correct equations of motion and S-matrix are recovered, are also novel and much more general than previous results in the literature.

Going off-shell allows us to compute the string partition function on off-shell backgrounds, i.e. on a non-solution to the equations of motion (at least perturbatively in the off-shell variation). This implies that the worldsheet field theory is a *QFT* rather than a CFT. Given the importance of conformal invariance to the consistency of the standard formulation of string theory, you might reasonably think that this would make off-shell string theory inconsistent or ambiguous. But as we shall see this is not the case. There are two serious problems which need to be addressed in order for the formalism to be well-defined.

The first issue, which arises at arbitrary genus g, has to do with the need to arbitrarily fix a Weyl frame $\omega$ on the worldsheet. Fortunately, at the end of the day this arbitrary choice does not matter! As we shall show, this is because the effects of changing $\omega$ can be fully absorbed into field redefinitions of the target space fields. This corresponds to renormalization of the worldsheet QFT.

While Tseytlin's off-shell formalism can be used at arbitrary genus g, the treatment of the sphere diagram (genus-0) is particularly subtle. This is because of the existence of a noncompact conformal Killing symmetry group SL(2,$\mathbb{C}$). This leads to the question of what it means to mod out by this group, when perturbing the worldsheet theory by vertex operators associated with non-conformally invariant sources. For this we need to make use of a special sphere prescription.

Tseytlin does *not* deal with the SL(2,$\mathbb{C}$) Möbius group by fixing 3 points, as this prescription does not properly extend to the off-shell case. Instead, at the $n$-th order of perturbation theory, he integrates *all n* vertex operators over the sphere to obtain a correlator $K_{0,n}$. This introduces log divergences as $n-1$ points come together on the sphere. To obtain the correct spherical string amplitude $Z_0$ for a sphere, Tseytlin then *differentiates* by the log of the UV cutoff $\epsilon$, so that (up to a determinable multiplicative factor) we get: [9]

$$Z_0 = \frac{\partial}{\partial \log \epsilon} K_0 \,, \tag{T1}$$

where $K_0 = \sum_n K_{0,n}$. We call this formula **T1** because it was Tseytlin's *first* sphere prescription, and also because it involves *one* derivative with respect to the RG flow.

By taking the QFT to be a nonlinear sigma model, Tseytlin checked that this prescription gives good answers for the first few terms in the effective action $I_0$, at least for massless fields of (super)string theory in the long wavelength regime where the characteristic radius of curvature of the target spacetime $r_c \gg l_s$ [10, 11].

Unfortunately, in bosonic string theory **T1** wrongly implies that there is a tree-level tadpole associated with the tachyon field. (This tadpole arises because the identity operator has a nonvanishing 1-point function on the sphere; hence the **T1** action is not extremized with respect to varying the tachyon zero mode, i.e. a cosmological constant.) To eliminate this tadpole, Tseytlin proposed a *second* sphere prescription involving *two* derivatives with respect

to the RG flow [12, 13]:

$$Z_0 = \left( \frac{\partial}{\partial \log \epsilon} + \frac{1}{2} \frac{\partial^2}{(\partial \log \epsilon)^2} \right) K_0 \,. \tag{T2}$$

Although at first this **T2** prescription looks very *ad hoc*, it actually has deep connections to the $c$-theorem which we will explain in what follows.

In this article, we will explain how to use these prescriptions to recover standard string theory results. The precise details depend on the size of the UV cutoff $\epsilon$ on the worldsheet. Adjusting $\epsilon$ (which is a special case of RG scheme dependence) controls the degree of nonlocality in target space [14]. In particular:

- In the limit where $\log \epsilon^{-1}$ is finite, we recover an approximately local action for light string fields. We will show that this action's equations of motion are satisfied *if and only if* the $\beta$ functions vanish (at least to all orders in perturbation theory).

- In the limit where $\log \epsilon^{-1} \to \infty$, we recover the standard Euclidean S-matrix. We will show that in this regime, the effect of applying **T1** or **T2** is equivalent to *modding out by the gauge orbits* of SL(2,$\mathbb{C}$), acting on the positions of $n$ punctures. We also show how to recover the Lorentzian S-matrix by integrating over complex values of the UV cutoff $\log \epsilon^{-1}$, in which case we obtain Witten's $i\varepsilon$ prescription for the internal poles [15].[2]

Another important difference between these two regimes, is the allowed dimensions of the vertex operator perturbations. In the S-matrix regime it only makes sense to perturb the worldsheet by marginal primaries, since strings that propagate out to infinity always obey the mass-shell condition.

On the other hand, when deriving the local action, we allow for the worldsheet Lagrangian to be perturbed by non-(1,1) operators, so long as their operator dimension lies within a window which we call the "renormalizability condition", which is more restrictive for **T1** than **T2**. If $n$ is the order of perturbation theory (e.g. when calculating an $n$-point function perturbing away from an on-shell string background), then the dimension $\Delta = h + \bar{h}$ must satisfy:

$$\textbf{T1}: \qquad 2 - 2/n < \Delta < 2 + 2/n \,, \tag{1}$$

$$\textbf{T2}: \qquad 0 \le \Delta < 2 + 2/n \,. \tag{2}$$

If these conditions are not satisfied then there are unwanted tadpole terms in the action which cannot be dealt with by the sphere prescription in question.[3] These conditions are weaker than those imposed by Tseytlin, who usually only works in an $\alpha'$ expansion, which corresponds to perturbation theory in $\Delta - 2$.

Vertex operators satisfying these conditions can be either primaries $\mathcal{P}$, or else non-minimal curvature (dilaton-like) terms of the form $R\mathcal{P}$. Strangely, the latter terms have the opposite sign in the action, a necessary consistency condition for recovering the "conformal mode problem" [16] of the action of general relativity. (Of course it would be an even bigger problem if GR could not be recovered from the low energy limit of string theory, so we regard this opposite sign as a good thing!)

Not surprisingly, there is also a close connection between the string action and the c-theorem for 2d QFT [17]. (It would be too shocking of a coincidence if there were two conceptually unrelated functionals of 2d QFTs that are extremized only by CFTs.) The relationship

---

[2]As noted in section 5.7, the precise value of the amplitude at the poles is somewhat scheme dependent, so we cannot check the **T1** prescription by simply looking at the numerical coefficients of a $\log \epsilon$ expansion when sitting directly on the poles.

[3]It is possible that these restrictions could be evaded with some prescription more general than **T2**, but we leave this to future work.

between the action and C-functions is a little subtle. In addition to reviewing Tseytlin's important contributions to this subject, we will also provide a novel connection between the trace formula for **T2** prescription and planar c-theorems.

**Background material.**   The quest to derive a low-energy effective action for strings (perturbatively in $\alpha'$) is almost as old as string theory itself [18]. The history and development of the nonlinear sigma model (NLSM) and its connections to string theory are rich and predates Tseytlin's work [19–31]. Of particular relevance to our work are [32–40].[4] The work of [41] and [42] inspired the **T1** prescription of Tseytlin [9].[5] The relationship between the work of Curci and Pafutti [43] and our work will be pointed in section 7. A relatively recent work that studied the UV divergence structure of nonlinear sigma models in closed and open string theories can be found in [44, 45].

Among Tseytlin's numerous works on off-shell string theory, the ones most important to us are the following: The central charge action [46, 47] (which shows the relationship to the c-theorem [17]), and the connection to Weyl invariance, the trace anomaly, and beta functions, which were derived in [48, 49] for the massless modes of the closed bosonic string. Issues related to Weyl ambiguities, RG schemes and field redefinitions of the off-shell string effective action were discussed in [50]. The detailed calculation of the NLSM partition function on a compact worldsheet were presented in [51–53]. The question of how to extend the formalism to the bosonic string tachyon was discussed in [54], and the formalism was extended to cover it in [12] by introducing the **T2** prescription.[6]

The sigma model approach is most successful only when the characteristic length of the background spacetime is much greater than the string scale, $l_s = \alpha'$. A major limitation of the sigma model approach to string theory is ability to handle only (nearly) *renormalizable* interactions within conformal perturbation theory. In the space of of two-dimensional quantum field theories, this limitation means that the RG space is limited to massless and tachyonic perturbations. Yet, the full string dynamics include relevant and massive deformations. A systematic attempt to address this shortcoming can be seen in the work of [35, 72–75] by interpreting the equations of of motions of the strings as the *exact* renormalization group equation [76, 77]. We will discuss this limitation in light of the **T1** and **T2** prescriptions in section 6.2. However, even this attempt to include nonperturbative dynamics of the string itself suffers from its own shortcomings [78].

We assume the reader is already familiar with the standard presentation of string theory at the level of [10], and we also refer to Schwinger methods both in field theory [79] and for the string propagator [80]. We also use extensively some basic facts about renormalization theory for local QFTs [81, 82]: namely (i) that the different choices of RG scheme are equivalent to smooth coordinate changes on the space of couplings; and (ii) while the coefficients of log divergences are universal, it is always possible to find an RG scheme which eliminates all

---

[4]The relationship of the Möbius *extra leg* logarithmic divergence to the *S*-matrix was discussed in [36].

[5]In [41], the regularized volume of the SL(2,$\mathbb{C}$) group was calculated and was pointed out that, it is infrared divergences like all divergences in string theory. In [42], the divergence coming from the limit of $n-1$ vertex operators colliding on the torus was considered. Quantum mass renormalization was used to absorb the divergence. This factorization channel of an $n$-point function is the one that Tseytlin used later to factorize the external leg pole from an $n$-point amplitude before fixing 3 point on the sphere. This limit and the tadpole limit, where all $n$ vertices collide are central in our discussion in this paper.

[6]Although, in this paper, we primarily focus on the subtleties of computing the sphere (tree-level) partition function in closed bosonic string theory [9], Tseytlin's off-shell formalism [3–5] has wider applicability and extends to genus $g > 0$ topologies [55–59], open strings [60–63, 63–66], supersymmetric effective actions [4, 64, 67], and the inclusion of curvature-cubed terms in the string effective actions [68]. There are also explicit computations showing the equivalence, on-shell to order $\alpha'$, of the closed bosonic string equations of motion (for the massless modes) to the vanishing of the 2-loop beta functions [61, 69]. The relationship of Tseytlin's off-shell nonlinear sigma model first-quantized formalism to string field theory was discussed in [70, 70, 71].

power law divergences (and convergences) in the UV cutoff $\epsilon$.[7]

In general we will not be very careful to keep track of the many positive multiplicative constants which arise for the worldsheet sphere partition function $Z_0$. It is not usually very valuable to worry about them because, in any calculation which involves only sphere and torus diagrams, such factors can be absorbed into a rescaling of the dilaton. But for a sufficiently complex calculation involving additional genera, it would certainly be necessary to derive the correct numerical factor to place in front of the sphere partition function. Our paper contains enough information to indicate in principle how this factor can be computed, modulo target space field redefinitions.

We also do not use the BRST formalism in this article, but content ourselves with the use of the Faddeev-Popov trick while treating the zero modes specially. We hope in future work to more carefully explore the BRST anomalies that appear on the off-shell worldsheet.[8] We also hope to make more solid connections to string field theory—although there are scattered references (especially in section 4) in this paper indicating how we believe these two off-shell formalisms are related to each other.

**Plan of paper.** The outline of this paper is as follows: in section 2 we review gauge-fixing in the on-shell formalism, and discuss why the sphere partition function vanishes on-shell. In section 3 we describe how to perform an equivalent gauge-fixing of the off-shell string, and we explain why the choice of Weyl frame does not result in any problematic ambiguities. We also introduce the perturbative expansion of the string amplitude. In section 4 we highlight the important role of the UV cutoff $\epsilon$ on the worldsheet and explain how it acts as an IR cutoff on the string propagator, controlling the degree of locality in the off-shell theory.

In sections 5 and 6 we prove that Tseytlin's sphere prescription gives the right answers for the tree-level S-matrix and equations of motion respectively, to all orders in perturbation theory, when expanding around an arbitrary worldsheet CFT. We also indicate the regime of validity of the **T1** and **T2** prescriptions in terms of the operator dimensions of off-shell perturbations (which includes all orders in $\alpha'$). In 7 we explain the close relationship between Tseytlin's action and c-theorems on the sphere and plane.

In part II of this work [7], building on the formal explanation of Tseytlin's off-shell formalism in this paper, we will provide a more explicit derivation of the Einstein-Hilbert action from the worldsheet sigma model. This allows us to explain the Susskind-Uglum calculation of classical black hole entropy from off-shell closed string theory, using the formula

$$S = (1 - \beta \, \partial_\beta) Z_0 \big|_{\beta = 2\pi}, \tag{3}$$

where $\beta$ is the conical opening angle. We will tentatively make some first steps towards making sense of the S&U open string picture. We will also compare the S&U to a rival method for calculating black hole entropy by analytically continuing (on-shell) $\mathbb{Z}_N$ orbifolds. Unfortunately, this method does not give the correct entropy unless—as seems promising, following Dabholkar [83]—we allow tachyons to condense on the orbifold. Finally, we will conclude by suggesting possible avenues for further calculations of entropy in the off-shell formalism, including the bulk side of holographic AdS/CFT spacetimes.

**Index conventions.** In order not to make any presuppositions about the target space field content of the string theory background, wherever possible we use Zamolodchikov-style index

---

[7]Examples of RG schemes which do this automatically are $\zeta$-function or dimensional regularization, or more generally any technique involving analytic continuation from a convergent region. In other regulator systems (such as the heat kernel or hard disk), one may simply cancel these power laws divergences and convergences by hand, which can always be done without introducing a new dimensional scale.

[8]We believe these BRST anomalies play an important role in ensuring the tree-level S-matrix is nonzero even though in Tseytlin's method we integrate all $n$ vertex operators.

notation for the coupling constants of the worldsheet theory (although in section 6 the curvature couplings associated with the dilaton require special treatment). However, for those who like to see more concrete expressions, we do write out explicitly the sphere partition function and action for the NLSM graviton and dilaton in section 7.4, which will be more carefully derived in part II of this work [7].

Our index conventions throughout both papers are:

| | |
|---|---|
| $\mu$ | tangent index (target space), |
| $A$ | tangent index (worldsheet), |
| $\aleph$ | species (including polarization), |
| $a$ | target space mode (incl. species), |
| $i$ | like $a$ but primary modes only, |
| $Ri$ | primary $i$ times Ricci scalar, |
| $m$ | worldsheet mode. |

For all but the last of these, there is a distinction between upstairs and downstairs indices; for example a field is $\phi^a$ but its equation of motion is $E_a$. The Einstein summation convention is sometimes used, but not in certain expressions where it might be confusing.

In some cases, e.g. for the S-matrix, $i$ is implicitly restricted by context to *marginal* primaries (modulo pure gauge modes[9]). We also use the notation $\tilde{\Phi}^i := \phi^{Ri}$ for non-primary (dilaton-like) curvature couplings on the worldsheet.

For products of $n$ factors (where $n$ = the number of vertex operator insertions on the worldsheet), we do not use an index, but simply write $\prod^n$ and let the dependence of each factor on $1\dots n$ be implied.

## 2 The on-shell partition function

### 2.1 Gauge fixing

Recall that in bosonic string theory, the value of the on-shell partition function is

$$Z_{\text{on-sh}} = \int \frac{[\mathrm{d}X][\mathrm{d}g]}{\text{Diff} \times \text{Weyl}} \exp\left(-\int \mathrm{d}^2 z\, \mathcal{L}_{\text{CFT}}[X, g]\right), \tag{4}$$

where $X$ is the target space coordinates, $g$ is the metric, and we are not yet including any vertex operator insertions.

Since the Diff and Weyl symmetries are noncompact, the vertical bar in this expression is better thought of as *quotienting out* the field space by the symmetry group, rather than dividing by a number. Doing this properly requires the specification of a covariant measure $[\mathrm{d}\xi][\mathrm{d}\omega]$ on the Diff × Weyl group. (The combination of measures $[\mathrm{d}g]/[\mathrm{d}\xi][\mathrm{d}\omega]$ is what gives the $c = -26$ conformal anomaly of string theory that needs to be cancelled by a suitable matter CFT with $c = +26$.)

Since we are on-shell, $\mathcal{L}_{\text{CFT}}$ is the Lagrangian of a conformally invariant theory. Technically, it is only the combination of the Lagrangian and the measure factors which needs to be Weyl invariant. That is, if we have an on-shell string background, the path integral taken as a whole is invariant under Diff × Weyl, and therefore it makes sense to mod out by this group.

---

[9]In Lorentzian signature, the null-propagating pure gauge modes are (oxymoronically) *primary descendants* [84]. There are also constraint modes that are non-primary non-descendants.

In order to do calculations it is convenient to gauge-fix to a family of metrics $g = \hat{g}(\tau)$, thus introducing the Faddeev-Popov determinant:

$$Z_{\text{on-sh}} = \int \frac{[\mathrm{d}\tau]}{\text{CKG}} \int \Delta_{\text{FP}}[\hat{g}(\tau)] Z_{\text{CFT}}[\hat{g}(\tau)] \,, \tag{5}$$

where $Z_{\text{CFT}}$ is the path integral over $X$ and there remains a finite-dimensional integral over conformal moduli $\tau$. If the choice of metric $\hat{g}$ does not fully fix the symmetry, we must still mod out by the conformal Killing group (CKG) of $\hat{g}$. As is well known, $\Delta_{\text{FP}}$ can be re-written in terms of the $b, c$ ghosts [10]:[10]

$$Z_{\text{on-sh}} = \int \frac{[\mathrm{d}X][\mathrm{d}\tau][\mathrm{d}b][\mathrm{d}c]}{\text{CKG}} \exp\left(-\int \mathrm{d}^2 z \, \mathcal{L}_{\text{CFT}}[X, b, c, \hat{g}(\tau)]\right). \tag{6}$$

That is,

$$Z_{\text{on-sh}} = \int \frac{[\mathrm{d}\tau]}{\text{CKG}} Z_{\text{ghost}}[\hat{g}(\tau)] Z_{\text{CFT}}[\hat{g}(\tau)]. \tag{7}$$

## 2.2 The sphere diagram vanishes on-shell

In the case of genus-0, the Teichmüller space of $\tau$'s is zero-dimensional, while the CKG group SL(2,$\mathbb{C}$) is noncompact, so naively we get $Z_0 = K_0/\infty = 0$, where $K_0$ is the genus-0 partition function without the CKG factor. Hence the tree-level (classical) string action $I_0$ vanishes!

Actually, this *is* the correct answer when the worldsheet is a CFT, and when there are no vertex operator insertions. This is because the classical string action

$$I_0 = -\int \mathrm{d}^D X \sqrt{G} e^{-2\Phi} \left[ 4\nabla^2 \Phi + R - \frac{1}{12} H_{\mu\nu\xi} H^{\mu\nu\xi} + O(\alpha') \right], \tag{8}$$

is (up to a total derivative) proportional to the dilaton $\Phi$'s equation of motion $E_\Phi$, and therefore vanishes on-shell, at least for a compact target space. For a noncompact target space, there can be boundary terms in the classical action, but it is unknown how to calculate these terms from a worldsheet perspective.[11]

This vanishing of the classical action on-shell has also been confirmed within the string field theory formalism [89] (again up to a boundary term) by using the ghost-dilaton theorem [90].

However, if we go off-shell, $I_0$ does not vanish, and so to calculate $I_0$ we need a prescription for dealing with the CKG off-shell.[12]

In the standard textbook approach to string theory, we first find the conditions for the beta functions to vanish: $\beta^a = 0$. Then we observe that, mysteriously, these are proportional to the equations of motion $E_a$ coming from an action like (8). This is unsatisfactory, as there ought to be a way to derive the classical string action directly from the sphere partition function. This is what is done in Tseytlin's approach.

---

[10]Because we did not gauge fix the CKG, there are no zero modes of the $b, c$ ghosts in the current formalism. However, the measure on the CKG (inherited from $[\mathrm{d}\xi][\mathrm{d}\omega]$) still provides the necessary "ghost zero mode" contribution to the calculation of the $c = -26$ anomaly. (This factor is conceptually distinct from issues related to the noncompactness of the sphere CKG.) A more advanced discussion involving BRST invariance would have to introduce these ghost zero modes and impose Siegel gauge [80, 85, 86].

[11]The boundary term cannot be determined by the usual $\beta$ function approach, which only gives the bulk equations of motion. To calculate it from the worldsheet we would need to find consistent target space boundary conditions for the closed string. For some recent progress calculating the sphere partition function in AdS$_3$ see [87], and for strings coupled to walls see [88].

[12]Another situation in which the sphere action does not vanish is noncritical string theory. For a calculation of the sphere partition function in Liouville theory, see [91]. Although this background can be equivalently expressed as an (on-shell) linear dilaton vacuum in critical string theory, that description reinterprets the Liouville mode as a new spatial dimension. So presumably the nonzero sphere action appears as a boundary term in that description.

# 3 Defining the partition function off-shell

## 3.1 Choice of Weyl frame

We now wish to consider strings (of general genus g) propagating in an off-shell background, for which the $\beta$ functions do not vanish. We therefore consider a Lagrangian $\mathcal{L}_{\text{QFT}}$ of some non-conformally invariant QFT, so that the theory depends on a choice of Weyl frame. If $\gamma_{ab}$ is a standard conformal metric in some coordinates, then a Weyl frame $\omega$ may be defined as a choice of $\omega(z)$ such that the QFT is coupled to a metric of the form:

$$g_{ab} \simeq e^{2\omega(z)}\gamma_{ab}(z), \tag{9}$$

where $\simeq$ means "up to diffeomorphism". (Since we only allow covariant worldsheet theories, this Diff ambiguity does not affect the value of any partition function that we consider.) We require the Weyl frame to be *covariant*, in the sense that it maps all elements of any equivalence class $\{\gamma\}/\text{Diff}$ into the *same* equivalence class $\{g\}/\text{Diff}$.[13]

For example, on a genus-0 worldsheet, we can pick $g_{ab}$ to be the standard uniform sphere metric at some specific radius $r$, where each choice of $r$ is a distinct Weyl frame. In this case $\{\gamma\}/\text{Diff}$ contains only a single element $\gamma_0$, and the definitions in the previous paragraph have been carefully phrased so that the SL(2,$\mathbb{C}$) symmetry of $\gamma_0$ does not prevent this Weyl frame from being considered covariant. In the higher genus case, $\{\gamma\}/\text{Diff}$ ranges over the usual worldsheet moduli parameters.[14] This is analogous to the gauge fixing of Weyl in the on-shell formalism.

We do *not* wish to treat $\omega$ as an extra dynamical scalar degree of freedom on the worldsheet (as is done in noncritical string theory in $D \neq 26$ dimensions) because this would spoil the QFT $\rightarrow$ CFT limit (e.g. if we take $\beta \rightarrow 2\pi$ in (3)). Instead, we will arbitrarily pick a *single* choice of Weyl frame $\omega$ for the worldsheet. (This sounds like a bad thing to do, but we will explain soon why it is acceptable!)

## 3.2 Off-shell gauge fixing

Next we wish to perform the same gauge-fixing steps as in part 2.1, but now in the case of an off-shell string theory (i.e. when the $\beta$ functions do not vanish).

We start with the off-shell bosonic partition function in the form:

$$
\begin{aligned}
Z_{\text{off-sh}}[\omega] &= \int \frac{[dX][d\gamma]}{\text{Diff}} \exp\left(-\int d^2z\, \mathcal{L}_{\text{QFT}}[X, g(\gamma, \omega)]\right) \\
&= \int \frac{[d\gamma]}{\text{Diff}} Z_{\text{QFT}}[\gamma, \omega],
\end{aligned}
\tag{10}
$$

where now $Z_{\text{off-sh}}$ depends explicitly on the particular choice of $\omega$. (In fact, $\omega$ appears not just explicitly in the Lagrangian, but also implicitly in the covariant definition of the measure factors.)

If $\mathcal{L}_{\text{QFT}}$ were the Lagrangian of a conformally invariant theory, $\omega$ would not make any difference and thus (10) would be equivalent to the usual unfixed partition function (4). In

---

[13]Since selecting any specific element of $\{g\}/\text{Diff}$ is coordinate-independent by construction, covariance does not place any substantive limits on what worldsheet geometries $g$ can be considered.

[14]Because every 2d conformal metric is locally indistinguishable, a covariant choice of $\omega(z)$ will inevitably depend on $\gamma_{ab}(z')$ at other points $z' \neq z$ on the worldsheet, breaking manifest worldsheet locality. This is not a big deal, as the usual way of gauge-fixing the Diff symmetry already sacrifices manifest locality (even on-shell), e.g. by treating zero modes separately. To make things more local, one could generalize the story to Weyl frames which depend on $X^\mu$, which could provide a useful way of going back to the Nambu-Goto formalism.

that case we could substitute

$$[\mathrm{d}\gamma] \to \frac{[\mathrm{d}g]}{\mathrm{Weyl}}. \tag{11}$$

However, since an off-shell theory is not conformally invariant, this substitution is invalid for $g = e^{2\omega}\gamma$ due to the integrand not being conformally invariant.

At this stage there are two ways to proceed. The most direct approach is to directly gauge fix the Diff symmetry alone in (10), which turns out to produce the same $b, c$ ghost sector as in the usual on-shell formalism [92]. Here we will follow a slightly more circuitous (but equivalent) route, that will allow us to follow the same steps as in the on-shell case.

In this approach we introduce a redundant Weyl parameter $\overline{\omega}$ such that $\overline{g}_{ab} = e^{2\overline{\omega}}\gamma_{ab}$. Here literally nothing depends on $\overline{\omega}$ (not even measure factors, which are still defined using $\omega$) so we can substitute:

$$[\mathrm{d}\gamma] \to \frac{[\mathrm{d}\overline{g}]}{\mathrm{Weyl}}, \tag{12}$$

where Weyl is now understood to act on $\overline{\omega}$ and not $\omega$.[15] The off-shell partition function now takes the form:

$$Z_{\mathrm{off\text{-}sh}}[\omega] = \int \frac{[\mathrm{d}\overline{g}]}{\mathrm{Diff} \times \mathrm{Weyl}} Z_{\mathrm{QFT}}[e^{2\omega}\gamma]. \tag{14}$$

Because this partition function exhibits Diff × Weyl invariance, just like the on-shell expression (4), we can now follow the same steps as for on-shell Faddeev-Popov gauge-fixing. This requires us to choose a $\overline{g} = e^{2\overline{\omega}}\hat{\gamma}$, and so we obtain:

$$
\begin{aligned}
Z_{\mathrm{off\text{-}sh}}[\omega] &= \int \frac{[\mathrm{d}\tau]}{\mathrm{CKG}} \int \Delta_{\mathrm{FP}}\big[e^{2\omega}\hat{\gamma}(\tau)\big] Z_{\mathrm{QFT}}\big[e^{2\omega}\hat{\gamma}(\tau)\big], \\
&= \int \frac{[\mathrm{d}\tau]}{\mathrm{CKG}} Z_{\mathrm{ghost}}[e^{2\omega}\hat{\gamma}(\tau)] Z_{\mathrm{QFT}}[e^{2\omega}\hat{\gamma}(\tau)].
\end{aligned}
\tag{15}
$$

Note that when we evaluated $\Delta_{\mathrm{FP}}$, we still obtained the usual $b, c$ ghost CFT, as is necessary for the QFT $\to$ CFT limit to be continuous. For the same reason, it is important that the moduli $\tau$ are restricted to the same "fundamental region" that we would have used in the conformally invariant case, due to the conformal invariance of $\gamma$.

What does it mean to mod out by the CKG in a theory which is not conformal?[16] When the genus g $\geq$ 1, this question is relatively simple because there we can always choose our gauge-fixed metric $\hat{g}$ so that the CKG acts on $\hat{g}$ as a normal Killing isometry. In this case, the symmetry preserves the UV cutoff $\epsilon$, and so the off-shell amplitude is simply given by the following amplitude:

$$A_{g,n} = \int [\mathrm{d}\tau] \frac{K_{g,n}}{\mathrm{Vol(KG)}} \qquad (g \geq 1), \tag{16}$$

where Vol(KG) is finite because the group of isometries is either compact (g = 1) or finite (g > 1). $K_{g,n}$ is the CFT correlation function for inserting $n$ vertex operators onto the genus g worldsheet. $K_{g,n}$ will be properly defined in (24).

---

[15]Equivalently, we may write this "fake conformal" partition function as

$$Z_{\mathrm{QFT}}[g] = \int \frac{[\mathrm{d}X][\mathrm{d}g]}{\mathrm{Diff} \times \mathrm{Weyl}} \exp\left(-\int \mathrm{d}^2z\, \tilde{\mathcal{L}}[X, g]\right), \tag{13}$$

where $\tilde{\mathcal{L}}$ is the unique Weyl-invariant functional which agrees with $\mathcal{L}_{\mathrm{QFT}}$ when our choice of Weyl frame is satisfied. In other words, $\tilde{\mathcal{L}} = \mathcal{L}_{\mathrm{QFT}}$ when $g_{ab} = e^{2\omega}\gamma_{ab}$. Here Weyl invariance uniquely defines $\tilde{\mathcal{L}}$ when $g_{ab} \neq e^{2\omega}\gamma_{ab}$, because a fully specified Weyl frame should pick out exactly one metric $g_{ab}$ in each Weyl orbit.

[16]Since the CKG factor involves Diff as well as Weyl, it remains present even in the approach where one gauge-fixes Diff without introducing the fake Weyl parameter $\overline{\omega}$, so it is not "fake".

But in the case of the sphere partition function $Z_0$, we still have to mod out by the noncompact CKG group SL(2,$\mathbb{C}$). This procedure will be postponed until section 6.7. The key point will be that the UV regulator $\epsilon$ actually cuts off the integral over the noncompact directions of the CKG rendering the partition function finite. As a result, the sphere effective action does not vanish in general when we go off-shell.

### 3.3 Weyl scheme and field redefinitions

So far, $\omega$ looks like an arbitrary choice, which breaks not only conformal invariance but also manifest locality on the worldsheet. How can we justify doing such a horrible thing? Fortunately, at the end of the day, this choice of Weyl frame actually does not matter! This is because the effects of picking a different $\omega$ can be fully absorbed into *field redefinitions* of the target space fields. From the worldsheet perspective, this is closely related to the process of RG flow, in which you can change the scale of the theory if you also change the coupling constants.

Recall that, for a general spacetime action $I[\phi(X)]$ (not necessarily coming from string theory), whose equations of motion for a given species $\aleph$ are $\delta I / \delta \phi^{\aleph}(X) = E_{\aleph}(X)$, the physics of the model is unchanged under an infinitesimal field redefinition $\delta \phi^{\aleph}(\phi)$. Under such redefinition, the action changes (up to a total derivative) as follows:

$$\delta I = \sum_a E_a \delta \phi^a = \sum_{\aleph} \int d^D X \, E_{\aleph} \, \delta \phi^{\aleph}, \tag{17}$$

where the $a$-index includes a sum not only over the species $\aleph$ but also over target space modes.

Note that, if $\delta \phi^{\aleph}(\phi)$ is local, then the resulting correction $\delta I$ is also a local functional of the fields (although this may no longer be true if one integrates $\delta \phi$ to get a finite field redefinition $\Delta \phi$). Furthermore, (17) also holds for quantum effective actions $I_{g \geq 1}$ if we take the expectation value of the right hand side.

It follows from (17) that if we add to the action $I$ any *perturbatively small* term which vanishes on-shell (i.e. is proportional to some $E_a$), the physics is equivalent to all orders in perturbation theory.

Now we let $\delta I$ be the effective action of target space string theory. If we consider two nearby Weyl frames $\omega$ and $\omega + \delta \omega$ on the worldsheet, the difference in their partition functions is proportional to the trace of the stress-tensor $T$, which can be written in terms of the beta functions $\beta^a$ of the local RG flow [39, 93, 94]:[17]

$$\frac{\delta}{\delta \omega(z)} Z[\omega] = \langle\!\langle T(z) \rangle\!\rangle_\omega = \sum_a \beta^a \langle\!\langle \mathcal{O}_a \rangle\!\rangle_\omega^{\text{string}}, \tag{18}$$

where $\mathcal{O}_a$ is the unintegrated worldsheet vertex operator

$$V_a = \int d^2 z \, \mathcal{O}_a(z). \tag{19}$$

Here the amplitude symbol $\langle\!\langle \cdot \rangle\!\rangle$ is like an expectation value, but without the division by $Z$. Hence, the resulting effective action $I$ is an *integral* over target space, rather than an average, allowing for noncompact geometries. By the notation "string" we mean that the full moduli space integral $\int [d\tau]/\text{CKG}$ is performed.

But in string theory the $\beta^a$'s are proportional to the equations of motion $E_a$. (A proof that this is true in the off-shell formalism, at least perturbatively, will be given in section 6,

---

[17]A typographical note: some works use $\overline{\beta}$ [49] for the local RG evolution, which differs by a total derivative from the $\beta$ [32] functions of a global dilation. But we omit the bar, as whenever the distinction matters we only use the local version (it does not matter when integrating $\beta^a \mathcal{O}_a$ over the whole worldsheet). Note also the difference in font from the inverse temperature $\beta$.

but in this section we take it as a premise.) So by integrating $\delta\omega$, we can show that the difference between any two Weyl frames is proportional to terms with $E_a$ in them.[18] It follows that, even off-shell, any two Weyl frames give equivalent results, up to a field redefinition.[19] From the *worldsheet* perspective, such field redefinitions correspond to scheme-dependence in renormalization theory.

If the difference between the Weyl frames is not infinitesimal, then we will have to integrate (18) to get a finite sized shift of the target space fields. For a sufficiently large Weyl transformation, this will generally resum to a *nonlocal* redefinition of target space fields. Hence, we expect to get an approximately local effective action, at distances larger than the string length $l_s$, only when the worldsheet in question is "reasonably compact"[20] (i.e. without long protrusions or handles), and when the UV cutoff $\epsilon$ is not very small compared to the characteristic size of the worldsheet.

From the perspective of *target space*, this nonlocal renormalization procedure represents a process in which the background fields $\phi^\aleph(X)$ at a given point $X$ are adjusted in order to take into account the effects of coherent waves of strings, propagating to $X$ from elsewhere in the spacetime. This will be explained further in section 4.

Please note, that at no point in this discussion do we allow worldsheet coupling constants (i.e. target space fields) of our worldsheet theory to depend on the position $z$, and hence $\beta^a$ is also independent of $z$. As this point is potentially quite confusing, let us compare explicitly the case of a uniform and non-uniform Weyl rescaling $\delta\omega$. If $\delta\omega = $ const. (independent of $z$, and also genus g and moduli $\tau$), then (18) tells us that (with summation implied, and using $Z = -I$):

$$\frac{\mathrm{d}Z}{\mathrm{d}\omega} = \beta^a \frac{\partial Z}{\partial \phi^a} = -\beta^a E_a \,, \tag{20}$$

and it can be seen by comparison to (17) that the necessary field redefinition is simply given by the RG flow $\beta^a$, without any need to use its proportionality to $E_a$.[21] But in the non-uniform case we have, at any g and $\tau$:

$$\frac{\delta Z_{g,\tau}}{\delta\omega(z)} = \beta^a \frac{\delta Z_{g,\tau}}{\delta\phi^a(z)} \,, \tag{21}$$

where, in this expression and the next, we allow $\phi^a$ to be an explicit function of $z$, simply to give a name to inserting the corresponding source into the worldsheet theory. However, in this case it is necessary to re-express $\beta^a$ in terms of $E_a$ to identify the appropriate $z$-independent field redefinition.[22] If, to be concrete, we suppose that there exists a Zamolodchikov-like metric $\kappa^{ab}$ for which there is a gradient flow $\beta^a = -\kappa^{ab}E_b$, then the required field redefinition is:

$$\delta\phi^b = -\kappa^{ab} \int_{g,\tau} \int \mathrm{d}^2z \, \frac{\delta Z_{g,\tau}}{\delta\phi^a(z)} \delta\omega(z) \,, \tag{22}$$

---

[18]Even on a higher genus g > 0 worldsheet, what matters for absorbing the leading order $O(g_s^{2g-2})$ Weyl ambiguities are the *tree-level* equations of motion. By virtue of the Fischler-Susskind mechanism [95, 96], one also expects corrections to the $\beta$ functions that are subleading in $g_s$. These subleading ambiguities presumably match up with the higher genus corrections to the equations of motion, but we do not consider that aspect carefully here.

[19]Since these field redefinitions are associated with $\beta$ functions (whose linear part vanishes for marginal primaries) they do not change the definitions of asymptotic particles in the S-matrix.

[20]We mean this phrase not in the technical topological sense, but in the sense used in discussions of gerrymandering political districts. *Topologically* noncompact worldsheets would be associated with nonlocality over *infinite* length scales.

[21]However, the fact that $\beta^a$ is itself proportional to $E_a$, implies that the field redefinition does not have any effects on-shell. The same does not necessarily apply in the non-uniform case.

[22]If we had instead tried to interpret $\delta I/\delta\phi^a(z)$ as a $z$-dependent equation of motion $E_a(z)$, then we would have needed to consider position-dependent couplings. But this temptation should be resisted as it would infinitely proliferate the number of target space fields, throwing the whole formalism into havoc.

where the LHS has no dependence on the dummy variable $z$. Hence, it is always fully possible to compensate for a *local* Weyl frame change with a *uniform* coupling constant redefinition. This redefinition is, therefore, quite distinct from the nonuniform coupling constant redefinition that would compensate for the change in the local RG formalism.

To recap, sections 3.2 and 3.3 have now shown it is possible to define string theory off-shell using worldsheet QFTs. There is a sense in which scale-invariance still plays an important consistency role, however it is the scale-invariance associated with the *renormalization group*, in which the beta functions need not vanish. Only at a fixed point (a worldsheet CFT) does this become scale-invariance of the worldsheet theory itself.

## 3.4 Conformal perturbation theory

We now consider an expansion of the worldsheet QFT in coupling constants $\phi^a$.

At least if we are perturbatively near a fixed point, any such QFT will be equivalent to a CFT coupled to a coherent gas of vertex operators sprinkled on the worldsheet. Let us write the QFT worldsheet Lagrangian as:

$$\mathcal{L} = \mathcal{L}_{\text{CFT}} + \sum_a \epsilon^{2(h_a - 1)} \phi^a \mathcal{O}_a \,, \tag{23}$$

where $\phi^a$ represents the components of a vector in coupling constant space, and $\epsilon$ is the UV cutoff.[23] In off-shell string theory, the vertex operators $\mathcal{O}_a$ do not have to be conformal, i.e. they are scalars of weight $(h, h)$ with $h = \bar{h}$ but possibly $h \neq 1$.[24]

If we are perturbing around a string background, the original CFT has vanishing central charge: $c = 0$.[25] (In this article, the term "CFT" will always implicitly include this condition unless we explicitly say otherwise.)

By doing perturbation theory in the bulk fields $\phi_a$, the effects of $\mathcal{L}_{\text{int}}$ at order $\phi^n$ involve evaluating the CFT correlation function for inserting $n$ vertex operators onto the genus g worldsheet:

$$K_g(\tau) = \sum_{n=0}^{\infty} K_{g,n}(\tau) \,, \tag{24}$$

$$K_{g,n} = \sum_{a_1 \dots a_n} \left( \prod^n \epsilon^{2(h_a - 1)} \phi^a \right) \frac{1}{n!} \langle\!\langle V_{a_1} \dots V_{a_n} \rangle\!\rangle_{\text{CFT}} \,, \tag{25}$$

where *all $n$* vertex operators $V_a$ are integrated even for g = 0. When perturbing around flat spacetime, the external leg vertex operators can be written in a momentum basis:

$$\mathcal{O}_a \sim \exp(i P_\mu X^\mu) \mathcal{O}_\aleph \,, \tag{26}$$

where $P_\mu$ corresponds to the momentum of an external particle leg and $\mathcal{O}_\aleph$ is an ($X^\mu$-translation independent) species operator. If $\mathcal{O}_a$ is a (1,1) primary, then it corresponds to a physical state obeying the on-shell condition $P^2 + M^2 = 0$, while scalar primaries of other weights correspond to *off-shell* external particle legs.

In order to regulate UV divergences when subsets of these vertex operators approach one other, we may introduce a *hard disk* of radius $\epsilon$ around each vertex operator insertion $\mathcal{O}_i(z)$, and then forbid these disks from intersecting each other (i.e. the proper distance of the vertex operators must be at least $2\epsilon$). See Fig. 1.

---

[23]We include this power of the UV cutoff so that the coupling constants $\phi^i$ are formally dimensionless on the worldsheet, as is usually done when defining the renormalization group flow. Since $\epsilon$ has units of length, its size is controlled by the Weyl frame $\omega$.

[24]Even if the couplings are marginal at linear order, they can still have nonzero $\beta$ functions at higher orders—in the off-shell approach, this is related to the fact that there are nontrivial interactions at tree level.

[25]If the target space is noncompact, the modes $i$ may become continuous, in which case $\sum_i$ may become an integral.

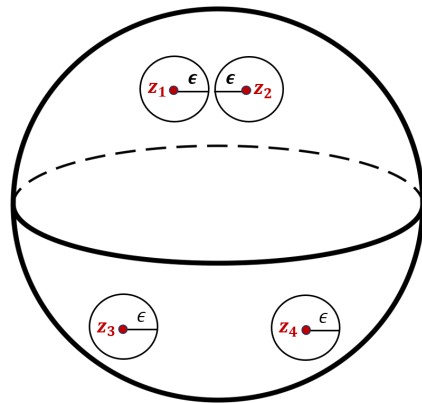

Figure 1: Hard disks of radius $\epsilon$ around each of four vertex operator insertions. These disks are not allowed to intersect, so the two disks on top are close to the boundary of the space of allowed positions.

### 3.5 The string amplitude

As the CFT correlator $K_{g,n}(\tau)$ is defined on a fixed background, it does not yet include integration over the moduli $\tau$ or modding out by the CKG. For the case of genus-1 or higher, the string amplitude is thus given by

$$A_{g,n} = \int [\mathrm{d}\tau] \frac{K_{g,n}}{\mathrm{Vol(KG)}} \qquad (g \geq 1). \tag{27}$$

On the other hand, in the case of a sphere we must use one of Tseytlin's sphere prescriptions:

$$\mathbf{T1}: \qquad A_{0,n} = \left( \frac{\partial}{\partial \log \epsilon} \right) K_{0,n}, \tag{28}$$

or

$$\mathbf{T2}: \qquad A_{0,n} = \left( \frac{\partial}{\partial \log \epsilon} + \frac{1}{2} \frac{\partial^2}{(\partial \log \epsilon)^2} \right) K_{0,n}, \tag{29}$$

the reasons for which we will justify later in detail. By resumming these expressions in $n$ using $-I_0 = Z_0 = \sum_n A_{0,n}$, we can obtain an $n$-independent expression for the effective action from $\mathbf{T1}$:

$$I_0^{(\mathbf{T1})} = -\left( \frac{\partial}{\partial \log \epsilon} \right) K_0, \tag{30}$$

and similarly for $\mathbf{T2}$.

Since differentiating with respect to $\log \epsilon$ is equivalent to RG flow, we can also write these prescription in terms of $\beta$ functions. In the case of $\mathbf{T1}$ we have:

$$I_0^{\mathbf{T1}} = -\frac{\partial K_0}{\partial \log \epsilon} = \sum_a \beta^a \frac{\partial K_0}{\partial \phi^a}, \tag{31}$$

while for $\mathbf{T2}$ things are a bit more complicated:

$$\begin{aligned} I_0^{\mathbf{T2}} &= -\left( \frac{\partial}{\partial \log \epsilon} + \frac{1}{2} \frac{\partial^2}{(\partial \log \epsilon)^2} \right) K_0 \\ &= I_0^{\mathbf{T1}} + \frac{1}{2} \left( \sum_{a,b} \beta^a \beta^b \frac{\partial^2 K_0}{\partial \phi^a \partial \phi^b} + \beta^b \frac{\partial \beta^a}{\partial \phi^b} \frac{\partial K_0}{\partial \phi^a} \right). \end{aligned} \tag{32}$$

Since all terms are proportional to $\beta$ functions, we immediately verify the expected result from section 2.2 that the action vanishes on-shell.[26]

Up to now we have been thinking of these vertex operators as off-shell perturbations to the worldsheet field theory. But by the operator-state correspondence we can also think of them as *external lines* in our Feynman diagram, in which strings join onto the worldsheet. From this perspective, $A_{g,n}(P_1, \ldots, P_n)$ gives us the (connected) $g$-loop contribution to a stringy $n$-point correlation function with possibly off-shell momenta. Since the correlation function is off-shell, we can Fourier transform to obtain the corresponding off-shell string correlator $A_{g,n}(X_1, \ldots, X_n)$ at finite spacetime positions.

However, $A_{g,n}(X_1, \ldots, X_n)$ is not quite the same thing as the usual position space correlator. Instead, it is more more analogous to a *truncated* connected $n$-point correlator in which all the external propagator factors $1/P^2 + M^2$ have been removed.

To see this, recall that, in the limit where all the external lines go on-shell (i.e. for each leg, $P^2 + M^2 \to 0$), this amplitude is identical to the usual S-matrix. To return to the S-matrix, we simply multiply the amplitude by a delta function for each external leg:

$$S_{g,n} = A_{g,n} \prod^n \delta(P^2 + M^2), \tag{33}$$

which forces each external leg to be a (1,1) primary. We would the interpret modes of positive/negative frequency as incoming/outgoing strings respectively.[27]

This procedure is different, however, from the usual LSZ prescription for recovering the S-matrix from the standard N-point Green's function $G_n$. In the LSZ procedure, we have to i) truncate $G_n$ by removing the factor of $1/P^2 + M^2$ for each external line. Only after doing that can we ii) impose the factor of $\delta(P^2 + M^2)$ for each leg.

Hence, to recover the string Green's function $G_{g,n}$ for a given genus $g$, we would need to restore the pole for each propagator:

$$G_{g,n} = A_{g,n} \prod^n \frac{1}{P^2 + M^2}. \tag{34}$$

This limit implicitly defines $G_{g,n}$, but only in the limit where $G_{g,n}$ is dominated by its external line poles. A more complete off-shell definition of $G_{g,n}$—which would require "sewing an external propagator" on to the truncated propagator—would likely require a picture more like string field theory.[28]

If we consider Tseytlin's amplitude $A_{g,n}$ for arbitrary off-shell momentum, it can exhibit some strange behavior. For example, in the 4-point tree amplitude $A_{0,4}$, the location of the *internal* pole of $A_{0,4}$ can get shifted away $P^2 + M^2 = 0$! However, this effect arises, not because of any physics associated with the internal legs, but because of the peculiar way in which the UV regulator $\epsilon$ on the worldsheet truncates the external lines. (Specifically, it truncates the external lines at a nonzero value $s$ of their Schwinger parameter, where $s$ depends in a holistic way on the rest of the diagram.) This will be described further in sections 4.2 and 5.5.

## 3.6 Two point amplitude

In the above formulae, the case $A_{0,2}$ calls for special attention.

---

[26]These expressions are manifestly local in RG space. So if e.g. you expand the action around two different nearby CFT's, you don't have to worry that the results will depend on which CFT you take as your initial background.

[27](33) is somewhat schematic, as in general, for fields with spin, there are spacetime indices in the propagator and hence in the on-shell condition.

[28]A better way to construct e.g. $G_{0,n}$ would be by cutting $n$ disks out of a genus-0 worldsheet, allowing each disk to be of arbitrary radius $r$ (thus allowing the external legs to be of arbitrary Schwinger length), and finally integrating over modular parameters $\tau$ and modding out by SL(2,$\mathbb{C}$).

The application of (34) to $n = 2$ primaries requires that $A_{0,2}$ look like a 2-point function times *two* distinct factors of $P^2 + M^2$, one for each external endpoint. As we will confirm by explicit calculation in section 6.3, $A_{0,2}$ therefore takes the form

$$-A_{0,2} \propto P^2 + M^2 \propto \Delta - 2, \tag{35}$$

which gives us a propagator term—schematically of the form $\phi(M^2 - \nabla^2)\phi$—in the effective action $I_0$, *without* the usual inverse. Here $\Delta = 2h$ is the operator dimension of the corresponding CFT vertex operator. Hence, the quadratic term of the sphere effective action is negative for relevant perturbations and positive for irrelevant perturbations, just like a c-function [17].

Since this formula vanishes for marginal perturbations with $P^2 + M^2 = 0$, one might think that the 2-point function in the S-matrix should also vanish. But this is not so, because in (33) there are also *two* powers of the delta function $\delta(P^2 + M^2)$. Hence $S_{0,2}$ has the indeterminate form $0 \times \infty$. A more careful analysis [97] of this situation (which also arises for particles) gives us the trivial delta function term in the connected S-matrix:

$$\delta^D(P_\mu^{\text{in}} - P_\mu^{\text{out}})\delta(P^2 + M^2), \tag{36}$$

in which a single string comes in and out without interacting with anything else. The normalization of this trivial term, while subtle to calculate from a worldsheet perspective, obviously has to be 1 by unitarity of the S-matrix.

## 3.7 String tadpoles

If, instead of expanding around a CFT, we choose to expand around an off-shell background that violates the classical equations of motion, then we will find that $A_{0,1} \neq 0$. This implies[29] that the off-shell background has an amplitude to emit *string tadpoles* and hence is unstable at linear order.

So long as we maintain a conceptual separation between the background spacetime and the strings propagating on it, this does not necessarily result in any inconsistency in the off-shell description.[30] However, because coherent states of string fields are equivalent to shifts in the background fields of string theory, one might think that on physical grounds, the effect of these emitted strings should be resummed in a way that effectively pushes the background into some nearby *on-shell* solution, from the perspective of test strings probing the situation.

There is a sense in which this can be true, but the precise requirements are subtle. It is certainly not true if we keep the UV cutoff $\epsilon$ at a fixed and finite value. But if we take a limit in which $\epsilon \to 0$, and then RG flow from there to some fixed scale $\mu$, the physics at $\mu$ will be governed by the IR limit of the theory defined at $\epsilon$. Now *if* this IR limit corresponds to an on-shell string background (which is plausible if we start near an on-shell background without tachyons) then the off-shell physics is equivalent to an on-shell scenario. A specific example of such a flow on a conical background will be described in part II of this work [7].[31]

Please note that these physical string tadpoles in $A_{0,1}$ should not be confused with the tadpoles in the sphere 1-point correlator $K_{0,1}$. These unphysical dilaton and tachyon tadpoles

---

[29]Assuming there are no compensating higher genus correction from $A_{g,1}$ effects, via the Fischler-Susskind mechanism [95, 96].

[30]There are arguments in e.g. [11] that off-shell observables don't make sense in string theory, because string theory involves gravity and there are no truly local observables in a diffeomorphism-invariant theory of gravity. Whatever the merits of this argument may be for a nonperturbative formulation of string theory, it does not affect our current formalism since, even when we go off-shell, we are still (as in the on-shell formalism) doing perturbation theory of strings on a fixed background.

[31]A more traditional form of renormalization would be to hold the physics fixed at $\mu$ rather than $\epsilon$ when taking the $\epsilon \to 0$ limit. This type of renormalization could be used to take a continuum limit of the worldsheet theory while remaining off-shell, but it is only consistent if the off-shell background flows to a UV fixed point (or an otherwise UV safe scenario).

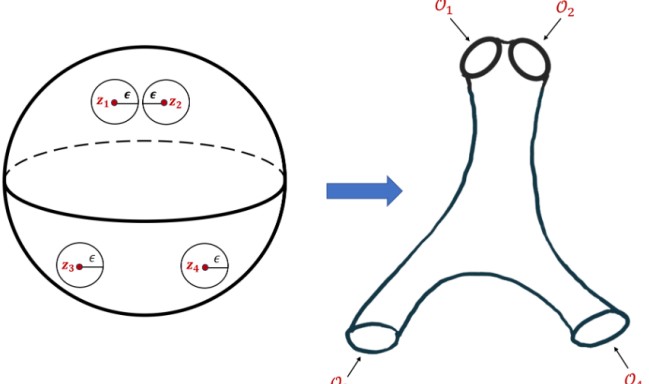

Figure 2: A conformal transformation which replaces the sphere regulated with a hard disk cutoff, with an open Riemann surface. The vertex operator insertions become state insertions on the boundaries. The length of the external and internal tubes is determined by the relative positioning of $z_1 \ldots z_4$ on the worldsheet, relative to our choice of Weyl frame $\omega$. In fact this is the sole effect of $\omega$ and $\epsilon$, as everything else is conformally invariant. In principle this conformal transformation allows one to re-express Tseytlin's off-shell formalism in the language of string field theory, but with an rather exotic rule for determining where to truncate the external propagators.

appear in $K_{0,1}$ even for CFTs. The purpose of Tseytlin's sphere prescriptions **T1** and **T2** is to eliminate these spurious tadpoles so that CFTs are solutions to the string equations of motion. How this works will be discussed in section 6.1.

# 4 Renormalization and propagating strings

If we want to understand the off-shell structure of string theory better, we'll need a good understanding of how UV divergences can appear on the worldsheet. In fact as we shall discuss in this section there are manifestations of such divergences even in scattering problems where the external particles are all marginal primaries.

## 4.1 Structure of divergences

Let us now discuss what kinds of UV divergences can appear on the worldsheet.

In the usual on-shell approach to string theory—where we allow only (1,1) insertions— there are 2 types of UV divergences which appear in $K_{g,n}$. These correspond to *separating* degenerations in which the worldsheet is divided into two pieces, by a single string propagator which becomes *long*. Such separating degenerations come in two kinds:

- *Momentum-dependent* divergences, which can occur for $n \geq 4$ when $n-2$ or fewer vertex operators approach each other. In this case one gets log divergences only at special values of the external momenta. This happens when the separating internal propagator satisfies its mass-shell condition.[32] In this case the degeneration is called *generic*, because for generic values of the momenta there is no log divergence.

---

[32]Confusingly, since log divergences produce $\beta$ functions that cannot be absorbed into a change of scheme, an internal propagator going *on-shell* is associated with the string equations of motion going *off-shell*. Expressed in terms of the target space field theory, a nonlinear term in the EOM can always be absorbed into field redefinitions unless it satisfies the *linearized* EOM, which corresponds to the propagator being on-shell.

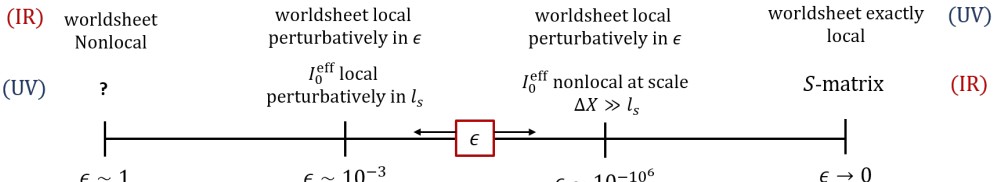

Figure 3: The sliding scale of worldsheet locality vs. target space locality, as controlled by the UV cutoff length $\epsilon$, with specific numerical values for illustrative purposes. (We take the Weyl frame to be a unit sphere.) Smaller values of $\epsilon$ make the worldsheet theory more local, but the target space effective action $I_0^{\text{eff}}$ becomes *less* local, ultimately culminating in the S-matrix regime where strings can make it out to asymptotic infinity. Since the nonlocality in target space grows very slowly as $\epsilon \to 0$, there is a wide range of values with good approximate locality on both sides. The large $\epsilon$ regime is confusing, but might be related to attempts to discretize the string worldsheet [98–101].

- *Momentum-independent* divergences, which can occur for genus $g \geq 1$ and $n \geq 3$ when $n$ or $n-1$ vertex operators approach each other on the worldsheet. These correspond to tadpole and mass renormalization effects, respectively [57,80].[33] Such degenerations are called *special*. In these cases a log divergence always exists whenever the external momenta satisfy the mass-shell condition.

In on-shell string theory, these two types of divergences need to be treated by totally different methods [15,80]. An important advantage of the off-shell approach is that both kinds of divergences can be treated on an equal footing, since even the so-called "momentum-independent" divergences can still be removed by taking the momentum of the external legs off-shell.

In both cases, we can break any amplitude into terms corresponding to different channels, such that in each channel there will exist some region in which the amplitude converges. Whenever an internal leg goes on-shell, this corresponds to a $\log \epsilon$ divergence. Analytically continuing to the *other* side of this pole then corresponds to throwing out power law divergences of the form $\epsilon^{-p}$, $p > 0$. Since divergences can always be absorbed into counterterms—and furthermore power law divergences can always be eliminated without introducing any additional scale into the worldsheet theory—we can simply strike out such powers of $\epsilon$ whenever they appear, when taking the $\epsilon \to 0$ limit.

In Tseytlin's formalism, because we don't fix 3 points, there are special degenerations (in which $n$ or $n-1$ vertex operators approach one another) in $K_{0,n}$ even at genus $g = 0$! These divergences are removed from $K_{0,n}$ by the **T1** or **T2** prescriptions.

Also, because the operators are taken off-shell, we can now handle the cases $n = 0, 1, 2$ in a manner which is homogeneous to the $n \geq 3$ cases. This is critical for the S&U paper because the classical black hole entropy [7] comes from the $n = 0, 1$ contribution to the sphere diagram, so if we can't handle these cases convincingly then we can't discuss the black hole entropy from a worldsheet perspective.

## 4.2   The regulated propagator

To describe such internal propagators on the worldsheet in a language more reminiscent of string field theory [6, 86], note that in the perturbation expansion defined above, the Weyl frame only matters in a neighborhood of size $\epsilon$ around each vertex operator insertion. Hence, we are still permitted to apply conformal transformations on the worldsheet minus the excised

---

[33]These divergences are associated with BRST anomalies that break gauge invariance of the S-matrix [80].

disks (see Fig. 2). In particular, we can convert any internal string propagator into a tube of radius $2\pi$, Euclidean Schwinger proper length $s$, and arbitrary twist $\alpha$.

Consider now a worldsheet that contains a single long degeneration in which one internal propagator leg becomes a long tube. Up to an $\epsilon$-independent additive constant $C$ which depends on the precise geometry of the worldsheet, the maximum possible tube length is given by $s_{\max} \approx 2\log\epsilon^{-1} = -2\log\epsilon$ (which happens when all vertex operators are bunched up in $O(\epsilon)$ sized clusters near 2 points $p$ and $q$ separated by an $O(1)$ distance on the worldsheet).

Hence, if there is a single long separating degeneration, the worldsheet amplitude will include a regulated propagator of the form:

$$\mathscr{P}_{\mathrm{reg}} = \delta_{L_0 - \bar{L}_0} \int_0^{2\log\epsilon^{-1} + C} \mathrm{d}s \exp\left(-s(L_0 + \bar{L}_0 - 2)\right) \tag{37}$$

$$\propto \delta_{L_0 - \bar{L}_0} \frac{1}{P^2 + M^2}\left(1 - \epsilon^{C\alpha'(P^2 + M^2)/2}\right). \tag{38}$$

Here we have used $L_0 + \bar{L}_0 - 2 = (\alpha'/4)(P^2 + M^2)$ and the constant $C$ depends on the precise details of how the tube is embedded in the Weyl-fixed worldsheet.[34]

The above integral assumes that there is a single long separating degeneration allowed on the worldsheet. In cases where there are multiple degenerations, we have to be more careful since the $\log\epsilon^{-1}$ instead controls the maximum length of certain *sums* of tube lengths on the worldsheet. We will treat this case more carefully in section 5.5.

## 4.3 Locality and the cutoff

Look again at (37). As usual in string theory, the UV regulator on the worldsheet plays the role of an IR regulator in target space. As pointed out by Susskind [14], the effective size of a string depends on the value of the UV cutoff $\epsilon$ [8], so sending $\epsilon \to 0$ allows the string to propagate long distances. Consider for example the genus-0 case (with the Weyl frame chosen to be a unit sphere), and let us see what happens to the Euclidean effective action $I_0^{\mathrm{eff}}$ as we adjust the value of $\epsilon$:[35]

- If $\log\epsilon^{-1} \sim 1$ then the internal tube will be cut off at short radius, and as a result the effective Euclidean action $I^{\mathrm{eff}}$ will be *local* over scales $\Delta X \gg l_s$. This is the regime in which we can derive a *local action* for string theory like (8).

- If $\log\epsilon^{-1} \gg 1$ (which means $\epsilon \lll 1$), the string can propagate for a longer distance, which means that the effective action $I^{\mathrm{eff}}$ becomes nonlocal over a somewhat longer scale $\Delta X \sim l_s \sqrt{\log\epsilon^{-1}}$ (due to massless propagators) or $\Delta X \sim M l_s \log\epsilon^{-1}$ (if there is a tachyon of mass $iM$).

- If we take the limit $\log\epsilon^{-1} \to \infty$, then strings can propagate over arbitrarily long distances. Then (37) becomes the standard propagator with a pole: $1/(P^2 + M^2)$. This is the *Euclidean S-matrix* regime.

See Fig. 3 for an illustration summarizing the effects that different values of $\epsilon$ have on locality at tree level.[36]

---

[34]Relatedly, the precise definition of a zero length tube ($s = 0$) is somewhat ambiguous unless, in the language of string field theory, we specify a *plumbing fixture*. This point is not important to us because here we are only concerned with the large $s$ aspects of the degeneration. The $\log\epsilon^{-1}$ cutoff on large values of $s$ is related to the *stub* in string field theory [86].

[35]See the discussion on p. 734-735 in [72] for the necessity of the UV cutoff in going off-shell.

[36]If we attempt to apply this same point of view to higher genus worldsheets, we run into the issue that they can also become nonlocal due to some modular parameter $\tau$ becoming large. In order for similar locality properties to

It should be noted that in the Euclidean S-matrix regime where $\epsilon \to 0$, it is problematic to introduce off-shell external lines with $h \neq 1$, due to the prefix factor in (25), because $\epsilon^{2(h_a-1)} \to \infty$ for a relevant operator, or $\to 0$ for an irrelevant operator.[37] So if you want to define an off-shell $n$-string correlator, this is best done at finite values of $\epsilon$. (In this paper, whenever we consider the S-matrix regime, we will also restrict to marginal primary operators.)

In the sections to follow, we will show that Tseytlin's sphere prescription gives good results at tree-level in both the S-matrix and local action regimes.

## 4.4  Lorentzian propagator and the $i\varepsilon$ prescription

The description of the Lorentzian S-matrix will be a bit more subtle as in this case we need to use the correct stringy $i\varepsilon$ prescription.

(Please note that we use the curly $\varepsilon$ symbol to refer to Feynman's $i\varepsilon$, and roman $\epsilon$ to refer to Tseytlin's UV cutoff. These are not equal, but it turns out they are closely related!)

According to Witten [15], one can derive a correct pole prescription for Lorentzian string theory as follows: we continue to treat the worldsheet metric as Euclidean, except in the case where there is a long tube opening up somewhere in the string worldsheet. For each such tube, we integrate $s$ along a contour for which the Schwinger time on the worldsheet eventually goes to positive *Lorentzian* infinity $t := -is \to +\infty$, rather than to Euclidean infinity. This produces an integral which is oscillatory in $t$ when $P^2 + M^2 \neq 0$, and constant otherwise. We regulate this integral with a small exponential damping factor of the form $e^{-\varepsilon t}$. Hence, the Lorentzian propagator is:

$$\mathscr{P}_{\mathrm{Lor}} = \delta_{L_0 - \bar{L}_0} \int_0^\infty \mathrm{d}t \, \exp\left(-it(L_0 + \bar{L}_0 - 2) - \varepsilon t\right) \tag{39}$$

$$= \delta_{L_0 - \bar{L}_0} \, \frac{-i}{P^2 + M^2 - i\varepsilon} \, . \tag{40}$$

Comparing the form of the Lorentzian propagator (39) to the regulated Euclidean propagator (37), we see that the $e^{-\varepsilon t}$ exponential damping factor can be obtained by performing an additional integral over imaginary values of $\log \epsilon^{-1}$:

$$\mathscr{P}_{\mathrm{Lor}} = -i\varepsilon \int_0^\infty \mathrm{d}t \, \exp(-\varepsilon t) \mathscr{P}_{\mathrm{reg}}(t), \quad \text{where} \quad t = -i\log(\epsilon^{-1}) + C, \tag{41}$$

and the value of the constant $C$ (which is related to the absolute value $|\epsilon|$ of the cutoff) is not important in the $\varepsilon \to 0$ limit.

In other words, the Lorentzian propagator can be obtained by taking $\log \epsilon^{-1}$ to be *imaginary*.[38] Then we integrate over an exponentially decaying distribution of $\log \epsilon^{-1}$ values, such that the characteristic size of $\log \epsilon^{-1} \sim i/\varepsilon$.[39] The $\varepsilon$ outside the integral ensures that the distribution is properly normalized since

$$\varepsilon \int_0^\infty \mathrm{d}t \, e^{-\varepsilon t} = 1 \, . \tag{42}$$

---

hold at loop level, it would be necessary to also cut off such modular parameters at some $O(\log \epsilon^{-1})$ value. (This could still be regarded as a UV cutoff in a Weyl frame where $\tau \to \infty$ corresponds to a degeneration in which a handle pinches off to a point.)

[37]A related problem affecting the off-shell *Lorentzian* S-matrix will be briefly discussed at the end of 5.7.

[38]Here we are assuming the RG flow is analytic, as it is in perturbation theory. Note that, since there can be divergences with non-integer powers as $\epsilon \to 0$, there is in general no requirement of periodicity when we take $\log \epsilon^{-1} \to \log \epsilon^{-1} + 2\pi i \mathbb{Z}$.

[39]But we cannot simply set $\log \epsilon^{-1} = i/\varepsilon$, as the integral over $\log \epsilon^{-1}$ is necessary to ensure convergence.

More generally, we propose that the Lorentzian tree-level[40] S-matrix can be obtained from Tseytlin's amplitude by the following relation:

$$S_{0,n}(\varepsilon) = \varepsilon \int_0^\infty \mathrm{d}t \, \exp(-\varepsilon t) A_{0,n}(t),  \qquad (43)$$

with $t$ defined as above,[41] or equivalently:

$$S_{0,n}(\varepsilon) = -i\varepsilon \int_0^{i\infty} \mathrm{d}(\log \epsilon^{-1}) \exp\left(i\varepsilon \log \epsilon^{-1}\right) A_{0,n}(\epsilon).  \qquad (44)$$

We will justify this odd looking rule in 5.4.

## 5 Obtaining the tree level S-matrix

Consider now the Euclidean S-matrix regime, where $n \geq 3$ and all the external legs are on-shell, i.e. perturbatively marginal $(1,1)$ primaries $\mathcal{P}$. From this we wish to show that we recover the usual S-matrix. After all, nobody is going to believe we have the correct *off-shell* prescription, unless it at least agrees with standard *on-shell* results! In this section, we show that this is indeed the case.

### 5.1 Gauge orbits of SL(2,$\mathbb{C}$)

First we give a general abstract argument for why Tseytlin's prescriptions should always work in the S-matrix context.

Let us define $\mathcal{PM}_{0,n}$ as the *pre-moduli* space of possible insertion positions $(z_1, \ldots, z_n)$ of vertex operators on the sphere, with all $z_i \neq z_j$. Note well that we have not yet modded out by SL(2,$\mathbb{C}$) so this space has $2n$ real dimensions. The usual on-shell moduli space would then be $\mathcal{M}_{0,n} := \mathcal{PM}_{0,n}/\mathrm{SL}(2,\mathbb{C})$ which has $2n-6$ real dimensions for $n \geq 3$. Our goal in this section is to define a regulated version of the moduli space, which takes into account the UV cutoff $\epsilon$. This is subtle because the UV cutoff is not conformally invariant. But it can still be done.[42]

First we define the *regulated* pre-moduli space $\mathcal{PM}_{0,n}^{(\epsilon)} \subset \mathcal{PM}_{0,n}$ as the subspace satisfying the condition that no two operator insertions are closer than $2\epsilon$ on the sphere.

Since in the S-matrix regime, the insertions are all marginal primaries, conformal symmetry guarantees that the CFT amplitude density

$$\mathrm{d}^{2n}z \, \langle\!\langle \mathcal{P}_{i_1}(z_1) \ldots \mathcal{P}_{i_n}(z_n) \rangle\!\rangle ,  \qquad (45)$$

is invariant under the action of SL(2,$\mathbb{C}$) acting on all points of $z_n$ simultaneously.

On the other hand, the cutoff prescription of the regulated pre-moduli space $\mathcal{PM}_{0,n}^{(\epsilon)}$ is not invariant under the conformal transformations in SL(2,$\mathbb{C}$), since the hard disk regulator

---

[40]To go beyond tree level, we would also need the right $i\varepsilon$ prescription for *nonseparating* degenerations contained within loop integrals. This requires cutting off modular integrals at $\tau \sim O(\log \epsilon^{-1})$ before applying (43).

[41]There is no $-i$ in (43), because any addition of a Lorentzian internal propagator also increases the number of Feynman vertices by 1, which introduces a compensating $i$ factor. The overall $i$ sign in the Lorentzian tree-level S-matrix comes from the Wick rotation of the $X^0$ temporal coordinate, which is already present in $A_{0,n}$ when it is evaluated in Lorentzian signature.

[42]There is an important difference between the off-shell Tseytlin's prescription and string field theory (SFT). In SFT, the local coordinate maps guarantee that the 3-point function even off-shell, when $\Delta_i \neq (1,1)$ is *always* truncated, which keeps dim $\mathcal{M}_{0,3} = 0$. Tseytlin's approach, on other hand, the truncation always happens *after* applying the **T1** or **T2** prescriptions. In fact, the pre-moduli space implies that the extra leg and tachyon tadpole logarithmic divergence *before* they are truncated, modify the fundamental tree-level 3-vertex $K_{0,3}$, such that dim $\mathcal{M}_{0,3} \neq 0$.

$$
\begin{array}{ccc}
\mathcal{PM}_{0,n} & \xrightarrow{\;\div\,\mathrm{SL}(2,\mathbb{C})\;} & \mathcal{M}_{0,n} \\[2mm]
\Big\downarrow \epsilon & & \Big\downarrow \epsilon \\[4mm]
\mathcal{PM}_{0,n}^{(\epsilon)} & \xrightarrow{\;\div\,\mathrm{SL}(2,\mathbb{C})\;} & \mathcal{M}_{0,n}^{(\epsilon)}
\end{array}
$$

Figure 4: A commutative diagram of the moduli spaces discussed in this section. The downward arrows represent UV regulation by the hard disk $\epsilon$, while the rightward arrows represent quotienting by the action of SL(2,$\mathbb{C}$). The arrow from $\mathcal{M}_{0,n}$ to $\mathcal{M}_{0,n}^{(\epsilon)}$ is implicitly defined by the other arrows.

$\epsilon$ explicitly refers to proper distance. Yet we may still quotient it by the action of SL(2,$\mathbb{C}$), by simply identifying any two elements of $\mathcal{PM}_{0,n}^{(\epsilon)}$ which are related by any element of SL(2,$\mathbb{C}$) acting on all insertions. We thus obtain a quotient space:[43]

$$
\mathcal{M}_{0,n}^{(\epsilon)} := \mathcal{PM}_{0,n}^{(\epsilon)} \,/\, \mathrm{SL}(2,\mathbb{C}) \,. \tag{46}
$$

We refer to elements of this space as gauge orbits $\Omega$.

From what we have said, it follows that the orbits included in the cutoff moduli space $\mathcal{M}_{0,n}^{(\epsilon)}$ are simply the subset of orbits of the unregulated moduli space $\mathcal{M}_{0,n}$ for which all insertions are separated by more than $2\epsilon$ in *at least one* SL(2,$\mathbb{C}$) frame. See Fig. 4.

It is tempting to try to compute the regulated volume Vol($\Omega$) of some representative gauge orbit, and then simply divide $K_0$ by that number. But this approach does not work, because the gauge orbits in $\mathcal{M}_{0,n}^{(\epsilon)}$ aren't all the same size with respect to the Haar measure on SL(2,$\mathbb{C}$)— their volume depends not only on $n$ but also (for $n > 3$) on the conformally invariant cross-ratios.

To correctly implement a division approach, we would have to calculate Vol($\Omega$) separately for each gauge orbit, which would be quite taxing. That is why it is so much easier to use Tseytlin's sphere prescriptions **T1** or **T2**, which—as we are about to show—are equivalent (in the S-matrix regime) to quotienting out by the gauge directions.

To demonstrate this, we first schematically calculate the volume of a given gauge orbit $\Omega$ of $\mathcal{M}_{0,n}^{(\epsilon)}$.[44] Since the cutoff $\epsilon$ is invariant with respect to the (compact) rotation group SU(2), the interesting contribution to Vol($\Omega$) comes from the regulated volume of the hyperbolic 3-space:

$$
\frac{\mathrm{SL}(2,\mathbb{C})}{\mathrm{SU}(2)} = H_3 \,, \tag{47}
$$

whose boundary is isomorphic to the worldsheet sphere, and which we take to have unit curvature radius. For $n \geq 3$, any sufficiently large boost of $H_3$ in any direction will push at least one pair of insertions closer than the cutoff distance (so all directions are regulated). See Fig. 5 for an image of the regulated gauge orbits in the cases $n = 2$ (which has Vol($\Omega$) = $\infty$) and $n = 3$ (which has finite volume). The former case is outside the scope of this section, but useful for gaining intuition about the geometry of gauge orbits.

---

[43]In other words, conformal symmetry is implemented as a *groupoid* rather than a group, because not every element $g \in \mathrm{SL}(2,\mathbb{C})$ is allowed to act on every element of $\mathcal{M}_{0,n}^{(\epsilon)}$.

[44]In the argument below, we adopt the convention that $\Omega$ is already defined (as an element of $\mathcal{M}_{0,n}$) independently of the value of $\epsilon$, although whether or not such an $\Omega$ is contained in $\mathcal{M}_{0,n}^{(\epsilon)}$ certainly *does* depend on $\epsilon$. This is important because we will eventually be differentiating with respect to $\log \epsilon$, and we need $\Omega$ itself to remain fixed.

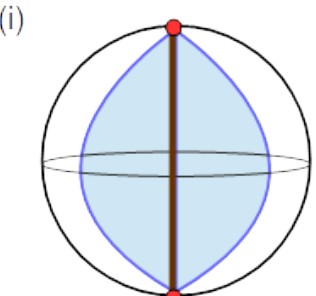
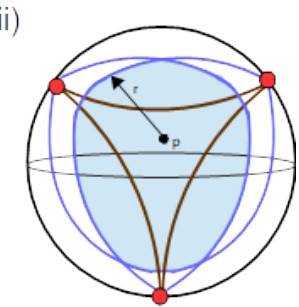

Figure 5: (i) A visualization of the regulated gauge orbit for the case $n = 2$. Two red vertex operators are shown on the $S_2$ spherical worldsheet, connected by a brown geodesic through the interior of hyperbolic space $H_3$, each point of which represents a conformal frame of $S_2$. The locus of light blue points, which is within a fixed $O(\log \epsilon^{-1})$ proper distance of the geodesic, are those points in the SL(2,$\mathbb{C}$) gauge orbit for which both points are at least $\epsilon$ apart on the sphere. Symmetry ensures that the surface of this locus is a fixed proper distance from the geodesic; hence the hyperbolic volume of the regulated gauge orbit is infinite. (ii) The regulated gauge orbit for three points ($n = 3$), i.e. $\mathcal{M}_{0,3}^{(\epsilon)}$. This is the set of points in the intersection (shaded light blue) of the three different $n = 2$ loci associated with each pair of points. (Only one dark blue edge is shown for each pair of vertex operators, as the edge on the other side is too far away to contribute to the boundary the regulated gauge orbit.) The volume of this gauge orbit (and any other regulated gauge orbit with $n \geq 3$) is finite, and can be integrated by picking a point $p$ somewhere in the interior, and shooting out rays $r$ in all possible directions. Although the boundary of the regulated orbit is not spherically symmetric or even smooth, all directions have the same universal $\log \epsilon$ contribution.

Let us call a gauge orbit $\Omega$ in the regulated space "large" if there exists any point $p \in \Omega$ which is hyperbolic distance $\gg 1$ from any of the cutoff boundaries, and "small" otherwise. We can calculate the volume of a large $\Omega$ by shooting out hyperbolic geodesics in all directions from $p$. On a given such ray $r$ with affine parameter $\lambda$, the regulated volume *per unit solid angle* is given by

$$
\int_0^{\log(a/\epsilon) + O(\epsilon^2)} d\lambda \, \sinh^2(\lambda) = \frac{a^2}{8} \epsilon^{-2} + \frac{1}{2} \log(\epsilon) + b + O(\epsilon^2), \tag{48}
$$

where $a \gg \epsilon$ for a large orbit. Here $a$ and $b$ (and the coefficients of further subleading terms) depend on the precise choice of $\Omega$, $p$ and $r$. However, the coefficient of the log divergence is universal. (This schematic form should be preserved when we do the solid angle integral over the space of all rays $r$ passing through $p$, so the coefficient of the log divergence is simply multiplied by $4\pi$, times the volume of SU(2).)

Hence (up to a multiplicative factor which is the same for all large orbits) the **T2** prescription gives us:

$$
\lim_{\epsilon \to 0} \left( \frac{\partial}{\partial \log \epsilon} + \frac{1}{2} \frac{\partial^2}{(\partial \log \epsilon)^2} \right) \mathrm{Vol}(\Omega) \propto 1. \tag{49}
$$

Since the volume of each large gauge orbit is counted as "1", the effect of **T2** is simply to mod out by the gauge symmetry.

Note that **T2** automatically kills the leading order cosmological constant divergence (or

anything else which scales like $\epsilon^{-2} = e^{-2\log\epsilon}$) because

$$\left(1 + \frac{1}{2}\frac{\partial}{\partial\log\epsilon}\right)e^{-2\log\epsilon} = 0\,. \tag{50}$$

That being said, in this S-matrix context, the simpler prescription **T1** is just as good, on the understanding that we are going to cancel out all power law divergences appearing in $K_{0,n}$. (Unlike the **T2** prescription, **T1** does not automatically eliminate the leading quadratic divergence of the cosmological constant from all $n$ points coming together.) But since we will need to cancel out power laws anyway to deal with poles coming from internal propagators, it is not a serious problem to do this by hand (or by means of the $i\varepsilon$ prescription that we will discuss later).

The discussion so far ignores the contribution of "small" gauge orbits, for which every valid SL(2,$\mathbb{C}$) frame has at least one pair of insertions whose proper distance is $O(\epsilon)$ (but $> 2\epsilon$) on the worldsheet sphere. Because small orbits are very close to being cut off by the regulator, we believe that their contribution to the partition function should be regarded as pure scheme; in particular they will not contribute to the coefficient of any log divergence.

**Open strings.** While our main concern in this paper is closed strings, the arguments in this section naturally generalize to open strings. Specifically, the genus-1/2 disk is invariant under a noncompact SL(2,$\mathbb{R}$) symmetry. Then we can make a similar argument involving the volume of 2d hyperbolic space $H_2$, whose ray-integral takes the form:

$$\int_0^{\log(a/\epsilon)+O(\epsilon^2)} \mathrm{d}\lambda\,\sinh(\lambda) = \frac{a}{2\epsilon} - 1 + O(\epsilon)\,. \tag{51}$$

Since this volume is not log divergent, the universal piece is the constant term. Hence, the disk analogue of **T2** is:[45]

$$A_{1/2} = \left(1 + \frac{\partial}{\partial\log\epsilon}\right)K_{1/2}\,, \tag{52}$$

which agrees with the earlier work of Witten [78, 102], while the analogue of **T1** (proposed by Polchinski and Liu [41]) drops the second term. See [103] for a detailed analysis of the role of SL(2,$\mathbb{R}$) in the disk partition function.

## 5.2 Generic momenta: Fixing 3 points

We now show that for generic values of the external momenta (i.e. when no internal propagators are on-shell) Tseytlin's prescription for the tree-level S-matrix agrees with the textbook method for treating the sphere, in which one gauge-fixes the position of 3 of the points. Let the tree-level amplitude defined by this method be $F_n$.

Let $\Omega_3$ be the $\epsilon$-regulated volume in the case of $n = 3$. In this case there is only one SL(2,$\mathbb{C}$) gauge orbit so the value of Vol($\Omega_3$) is uniquely specified. In this case, which always counts as "generic", fixing 3 points is obviously equivalent to modding out by $\Omega_3$.

Now we claim that, even for $n > 3$, conformal symmetry still implies that (up to scheme dependent terms):[46]

$$K_{0,n} = \mathrm{Vol}(\Omega_3)F_n\,. \tag{53}$$

---

[45] See the discussion in section 2 of [12].

[46] It was first pointed out in [41] that the volume of the SL(2,$\mathbb{C}$) group can be canceled out by the tree-level $n$-amplitude if we integrate over *all* vertex operator positions and then take the limit of all $n$ or $n-1$ points colliding. This directly implies that placing a UV cutoff of SL(2,$\mathbb{C}$) is equivalent (up to pure scheme) to regulating gauge orbits using a hard cutoff, as we do.

To see this, suppose we modify our regulator so that we place cutoff disks *only* around the 3 fixed insertions; thus allowing the other $n-3$ insertions to come arbitrarily close to each other, and/or to any one of the 3 special points. This could potentially introduce some unregulated divergences; but since these divergences involve at most $n-2$ vertex operators coming together—and because we are assuming generic external momenta—these divergences are pure power law, and thus can be eliminated by analytically continuing each such divergent channel to convergent regions. This defines $F_n$, which is finite.

We now integrate over the positions of the 3 special points, by acting with the SL(2,$\mathbb{C}$) symmetry on *all* $n$ insertions, wherever they are. This integral is cut off only when two of the 3 special points come together (regardless of the positions of the other points) so we get one extra factor of Vol($\Omega_3$). Using (49), we therefore find that for generic momenta, Tseytlin's amplitude is equivalent to fixing 3 points:[47]

$$A_{0,n} = F_n.\tag{54}$$

## 5.3 Generic momenta: Fixing 2 points

Another game we can play in the S-matrix regime is to fix the position of just 2 of the vertex operators on the sphere, e.g. we could pick one insertion to be at the North Pole and the other at the South Pole. This leaves unfixed the cylinder group $S_1 \times \mathbb{R}$. Note that this is still compatible with a special degeneration where $n-1$ points approach each other. When this happens, we will call the remaining point the *singleton*.

We need not discuss the twist generator $S_1$ in what follows, as its sole effect is to restrict our attention to scalars. But the $\mathbb{R}$ direction parametrizes the special degeneration—which is always a log divergence since all couplings involved are marginal. As before, we regulate this noncompact group with the cutoff $\epsilon$ to get an interval $\mathbb{R}^{(\epsilon)}$.

Assuming that the external momenta are generic, there is now exactly one power of $\log \epsilon$, coming from the integral over the regulated gauge orbit $\mathbb{R}^{(\epsilon)}$. If we start at one end of $\mathbb{R}^{(\epsilon)}$ (the North Pole) and integrate to the other end (the South Pole), the volume is given by

$$\text{Vol}(\mathbb{R}^{(\epsilon)}) = \int_0^{2\log(a/\epsilon)+O(\epsilon^2)} d\lambda = 2\log \epsilon^{-1} + O(1),\tag{55}$$

where $a$ depends on the details of the conformal cross-ratios, but only affects the $O(1)$ term.[48]

Comparing (55) to (48), we see that the log divergences in both are the same, up to a multiplicative factor (which happens to be negative!). Hence, acting on (48) with **T2** gives a result proportional to acting on (55) with either **T1** or **T2**. *It is therefore acceptable (up to a minus sign) to calculate $K_{0,n}$ in this regime where just $n-1$ points come together.*

The coefficient of the $\log \epsilon$ divergence is controlled by the $\beta$ function associated with $n-1$ points coming together at either pole. (Since it doesn't matter which we pick, let us say that the $n-1$ points come together at the South Pole while the singleton is at the North Pole.)

Hence, the $n$-point correlator (with 2 points fixed) takes the form (using the Einstein summation convention):

$$(K_{0,n})_{ij\dots z} = \log(\epsilon)\,\kappa_{ih}\,\frac{\partial^{n-1}\beta^h}{\partial\phi^j\dots\partial\phi^z} + O(1),\tag{56}$$

---

[47]To obtain the generic S-matrix, we could also simply divide $K_{0,n}$ by Vol($\Omega_3$), after removing the power laws and O(1) constants from both sides. But that might not work for non-generic momenta.

[48]There is an analogue for open strings if we fix 2 *boundary* operators on a disk. This reduces SL(2,$\mathbb{R}$) down to $\mathbb{R}$ and in the process introduces a new log divergence, which was not there before fixing the two points. We can then differentiate by $\log \epsilon$ to obtain the open string amplitude $A_{1/2,n}$ at generic momenta, and thereby relate the open string action to boundary $\beta$ functions.

and hence the amplitude may be written as:

$$\left(A_{0,n}\right)_{ij\ldots z} = \kappa_{ih} \frac{\partial^{n-1}\beta^h}{\partial\phi^j \ldots \partial\phi^z} \,, \tag{57}$$

where the Zamolodchikov metric $\kappa_{ij}$ is defined by the 2 point function of primaries inserted at both poles:

$$\kappa_{ij} := \langle\!\langle \mathcal{P}_i(z=0)\mathcal{P}_j(z=\infty)\rangle\!\rangle_{S_2}. \tag{58}$$

Note that although the $n$-point amplitude is symmetric in the modes $ij\ldots z$, the RHS does not look obviously symmetric. This is because, using the full SL(2,$\mathbb{C}$) gauge symmetry, we have the freedom to choose *any* of the $n$ points to be the singleton, while the others come together. In other words, Möbius symmetry guarantees that the RHS is symmetrical under permuting any pair of indices, e.g:

$$\kappa_{ih} \frac{\partial^{n-1}\beta^h}{\partial\phi^j \ldots \partial\phi^z} = \kappa_{jh} \frac{\partial^{n-1}\beta^h}{\partial\phi^i \ldots \partial\phi^z} \,. \tag{59}$$

It follows from the above that the tree-level effective action (only for marginal modes in the generic momentum regime) may be written as

$$I_0 = -\sum_{n=3}^{\infty} \frac{\kappa_{ij}}{n} \phi^i \beta^j_{(n-1)} \,, \tag{60}$$

where $\beta^j_{(n-1)}$ is the order $n-1$ beta function in the $\phi$'s.[49] Here, the factor of $1/n$ comes from symmetrizing over which field insertion is chosen to be the singleton. Even though we are in the S-matrix regime, the $\beta$ functions are still approximately local in target space because (in this subsection) we are staying away from internal poles in the S-matrix.[50] There is a compensating factor of $n$ when we differentiate the action (60) to obtain the (marginal primary part of) the equations of motion:

$$E_i = \partial_i I_0 = -\beta_i \,, \tag{61}$$

where in this expression we have lowered the beta function using the Zamolodchikov metric: $\beta_i := \kappa_{ij}\beta^j$.

This derivation of (61) uses the symmetry relations of the (59) of the beta functions in an essential way. If somebody simply presented you with the action in the form (60), and you didn't know that it came from conformally invariant amplitudes on the sphere, it would seem like magic that the correct equations of motion were obtained.

The demonstration of the corresponding result for *off-shell* variations must wait for section 6.

## 5.4 Non-generic momenta and $i\varepsilon$

The arguments in sections 5.2 and 5.3 fail if the momenta are not generic, because then it is possible to find log divergences which remain even *after* fixing 3 points. This makes the effects of the hard disk regulator more subtle, and in particular it is no longer possible to obtain the right answer simply by dividing $K_{0,n}$ by $\mathrm{Vol}(\Omega_3)$. (Similarly, after fixing 2 points, there are terms with more than one power of $\log\epsilon$ to worry about.)

Furthermore, in this case, fixing 3 points is *also* not the right on-shell prescription, because it does not treat all of the insertions symmetrically. Instead one may use e.g. the Deligne-Mumford construction [80, 104] in which the vertex insertions are held fixed and allows the

---

[49]The sum in (60) could also begin at $n=1$, since $\beta^j_{(0)}$ and $\beta^j_{(1)}$ vanish in the S-matrix regime.

[50]Since we are restricting $\phi^i$ to be marginal, the sum over $i$ implicitly includes a delta function $\delta(P^2 + M^2)$, but this does not produce a divergence because the resulting $\beta$ functions have support even away from $P^2 + M^2 = 0$, and are generically continuous with respect to taking $P^\mu$ off-shell.

worldsheet geometry to degenerate between them. Such degenerations can always be thought of as opening up long tubes inside the worldsheet.

Non-generic momenta can be important in tree-level scattering problems if the initial or final states are not momentum eigenstates. In such cases, one must integrate the S-matrix over a range of momenta. In this case, it is also necessary to have the correct $i\varepsilon$ prescription to deal with the poles which appear at special values of the momentum, as this provides a delta function contributions to the integrand.

As discussed in section 4.4, a correct prescription is to introduce a factor of[51]

$$\lim_{\varepsilon \to 0} \frac{1}{q - i\varepsilon}, \tag{62}$$

for each of the (at most $n-2$) internal propagators on the worldsheet, with Hamiltonian $q = L_0 + \bar{L}_0 - 2$. Please note that if we continue these amplitudes $q_1 \ldots q_{n-2}$ to complex values, the amplitude is holomorphic in the lower half plane of each $q$, since the pole has been pushed above the real axis to $q = i\varepsilon$.

In this section we show that the Tseytlin's sphere prescriptions **T1** or **T2** encode an $i\varepsilon$ prescription which is equivalent to the one above, if we translate between the two epsilons by integrating the UV cutoff along the contour proposed in (44):

$$S_{0,n}(\varepsilon) = -i\varepsilon \int_0^{i\infty} \mathrm{d}(\log \epsilon^{-1}) \exp\left(i\varepsilon \log \epsilon^{-1}\right) \frac{\partial K_{0,n}(\epsilon)}{\partial \log \epsilon}, \tag{63}$$

where the CFT correlator $K_{0,n}$ involves integrating $n$ vertex operators over all positions $z_1 \ldots z_n$, with a result that depends on the $q_1 \ldots q_{n-2}$. Each of these is associated with a Schwinger parameter $s_1 \ldots s_{n-2}$ whose minimum value is 0 and whose maximum value is *somehow* cut off by the $\log \epsilon^{-1}$ regulator. (We will explain exactly how this works in the next section, but suffice it to say for now that at finite $\log \epsilon^{-1}$ the maximum value of any $s$ is something of order $O(\log \epsilon^{-1})$).

The proof of equivalence is simple: just like (62), it turns out that (63) is *also* holomorphic in the lower half plane of each of the $q_1 \ldots q_{n-2}$ variables. To see this, note that when the $q$ values are all real, the contour going to $i\infty$ has an oscillatory integrand, because it is a Lorentzian signature Hamiltonian evolution. This is why the exponential damping factor $\exp\left(i\varepsilon \log \epsilon^{-1}\right)$ is introduced, to make the integral convergent. This exponential damping factor is sufficient because, without the damping factor, (44) is at worst power law divergent, with a maximum power of $(\log \epsilon)^{n-3}$ after differentiating by $\log \epsilon$.

If we now shift some of the $q$'s into the lower half plane, by (39) this only makes the integral even *more* convergent, so it follows that (63) converges throughout the lower half plane. Hence—since there can be no poles or branch points or other obstructions to analytic continuation—it must also be holomorphic in the lower half plane.

Furthermore, (62) and (63) agree away from any poles, because $A_{0,n}$ is insensitive to the details of the cutoff for generic values of the momenta. It follows that both (62) and (63) are each equivalent to a contour prescription in which one chooses to go around any poles on the real $q$ axis by deviating into the lower half-plane. Hence the are also equivalent to each other.

Note that it is very possible for two "equivalent" $i\varepsilon$ prescriptions to differ in their precise algebraic form at finite values of $\varepsilon$ (and indeed (62) and (63) do so differ). But any such equivalent prescriptions will give equivalent answers for the S-matrix whenever we do both of the following: (i) we must integrate over the (on-shell) external momenta using a continuous test function (ii) in the limit that $\varepsilon \to 0$. But there is no guarantee that two equivalent prescriptions give the same answer if you evaluate them exactly at a pole.

In section 5.6 we will encounter a concrete example of an S-matrix process contained in (63) which *vanishes* if conditions (i) and (ii) are met, but not otherwise.

---

[51]Regarding the absence of the usual factor of $-i$ in the numerator of (62), see footnote 41.

## 5.5 Correlators from fusion trees

Technically we've now completed our general argument that we recover the standard tree level S-matrix. But to see the way that $\epsilon$ cuts off tree level correlators more explicitly, we add the following observations. In general, a CFT sphere correlator can be calculated by means of *fusion trees* [84, 105, 106] which show how pairs of operators on the plane can be replaced with single operators, until at the end one has a 1-point function proportional to the identity operator. (Examples of such trees for $K_{0,4}$ will be shown in Fig 7.)

Hence, the fusion tree is a directed, rooted tree which ascends from $n$ nodes at the bottom layer of the tree (representing the $n$ vertex operator insertions on the worldsheet) up to a single node at the top layer (representing the identity). Each edge is associated with an operator $\mathcal{O}_i$.

The fusion tree can be regarded as a tensor network where each vertex represents an OPE fusion process:

$$\mathcal{O}_i(z_1)\mathcal{O}_j(z_2) \sim C_{ij}^k(z_1, z_2, z_3)\mathcal{O}_k(z_3). \tag{64}$$

We have not included in the above expression the scaling factor

$$(z_1 - z_2)^{-(h_i + h_j - h_k)}(\bar{z}_1 - \bar{z}_2)^{-(\bar{h}_i + \bar{h}_j - \bar{h}_k)}, \tag{65}$$

because each such factor can be reassigned to the corresponding edges in the graph. Taking into account also the rescaling of the measure factors $\mathrm{d}^2 z$ associated with each vertex operator insertion, one finds that each edge $e$ provides the following propagator factor:

$$\exp(-s_e q_e + i\alpha_e j_e), \tag{66}$$

where $s_e$ is a Schwinger time associated with a log of the change of scale, $\alpha_e$ is the twist, and $j_e = L_0 - \bar{L}_0$ is the angular momentum. Note that a marginal scalar has $q = 0, j = 0$ while the identity has $q = -2, j = 0$.

There are $n$ possible *chains* descending from the top of the tree to the bottom. For each such chain, the hard disk cutoff $\epsilon$ provides an upper bound on the *sum* of Schwinger parameters $s_a$ contained in each chain $\chi$:

$$\sum_{e \in \chi} s_e = \log(1/2\epsilon) + \omega(z_\chi), \tag{67}$$

where $\omega(z_\chi)$ is the Weyl factor of the vertex operator insertion at the base of the chain $\chi$. (The Weyl frame appears in this expression because the hard disk regulator $\epsilon$ refers to *proper* distance, and hence is not conformally invariant, even though all $n$ operator insertions are marginal. Hence, although it is easiest to calculate fusion trees on the plane, we have to remember that the $n$ point functions are actually regulated using the sphere metric with $e^\omega = 2/(1 + z\bar{z})$.)

To make this formula work properly we also need to include the Schwinger parameter $s_0$ of the identity operator at the top of the tree. Since this edge has only one endpoint, we arbitrarily define $s_0$ by comparison with the unit length $|z| = 1$. (This means that $s_0$ can be negative if there are operators separated by $|z_1 - z_2| > 1$).

Because the vertex operators at the base of the tree are $(1,1)$, $q = j = 0$ for the edges at the base of the tree, these edges do not contribute any factor to the amplitude (66).[52]

---

[52]This is on the assumption that we remain in the S-matrix regime. If we also take the external legs off-shell, then there would be an additional factor of $e^{-sq}$ associated with each of the external legs. Since by (67) the length of these external legs depends on the length of the internal legs, one finds that the poles appearing in $A_{0,n}$ (for $n \geq 4$) are strangely *shifted* away from the standard spectrum. But this is not for any reason having to do with the physics of the internal propagators being modified—it is simply an artifact of the strange way in which a spherical Weyl frame cuts off Feynman diagrams.

$$A_{0,4} \sim$$

Figure 6: The 4-point function and amplitude, with its 3 possible trivalent channels, plus an approximately local 4-valent process describing the physics away from the internal poles. In this section we focus on just a single trivalent channel. There is a $\log \epsilon$ divergence if one is sitting exactly on a pole.

It is therefore convenient to define a *truncated* chain $\chi'$ which excludes the bottom-most edge (which attaches to the vertex operator). Because the edges we just removed from the chain have positive Schwinger parameter $s > 0$, the truncated chains now satisfy an *inequality*:

$$\sum_{e \in \chi'} s_e < \log(1/2\epsilon) + \omega(z_{\chi'}). \tag{68}$$

Here $z_{\chi'}$ may be interpreted as the Weyl factor of whatever point the truncated chain would go to, if we take the limit that the attached vertex operators collide with each other.

Morally, these fusion trees look very similar to a tree level Feynman diagrams with $n$ external legs. But there are also some important differences:

1. The presence of a "tadpole" at the top of the diagram,[53] which in turn leads to:

2. The existence of an *orientation* in the tree proceeding away from the tadpole, and

3. One additional fake internal leg, arising as a result of one of the Feynman edges (which might be either internal or external) being bifurcated by where the tadpole joins onto the diagram. In the case where an external leg is bifurcated, the fake new leg is automatically on-shell.

As a result, to calculate an $n$-point integrated correlator $K_{0,n}$, then—in addition to the 2 center of mass degrees of freedom—we will also need to integrate over $(n-1)$ $s$-parameters and $(n-1)$ $\alpha$-parameters, rather than what we would expect on string field theory grounds, which is $(n-3)$ $s$-parameters and $(n-3)$ $\alpha$-parameters.

These 6 extra degrees of freedom are, of course, nothing other than our old friend the SL(2,$\mathbb{C}$) Möbius group, and arise because $K_{0,n}$ is defined by integrating over the (regulated) *pre*-moduli space $\mathcal{PM}_{0,n}^{(\epsilon)}$. Hence, by the arguments above, the effects of these extra degrees of freedom should be removed by imposing the **T1** or **T2** prescriptions. We will show how this works explicitly for $n = 4$ in the next section.

## 5.6 Example: 4 string scattering

In this section we will consider the simplest possible amplitude possessing nongeneric momenta, namely the 4-point amplitude $A_{0,4}$. In the $\log \epsilon^{-1} \to \infty$ limit, this amplitude contains a pole coming from the internal propagator.

---

[53]One might wonder why the diagrams are restricted to having only a single tadpole coming out of them. Usually, if a field theory allows tadpoles there can be any number. But this is just a feature of the sphere geometry being relatively compactified. Nothing stops you from considering a Weyl frame corresponding to a very blobby, nonuniform manifold with the topology of $S_2$, containing several tadpoles, if you really want to do that. In any case, the purpose of the sphere prescription was to eliminate the tadpole.

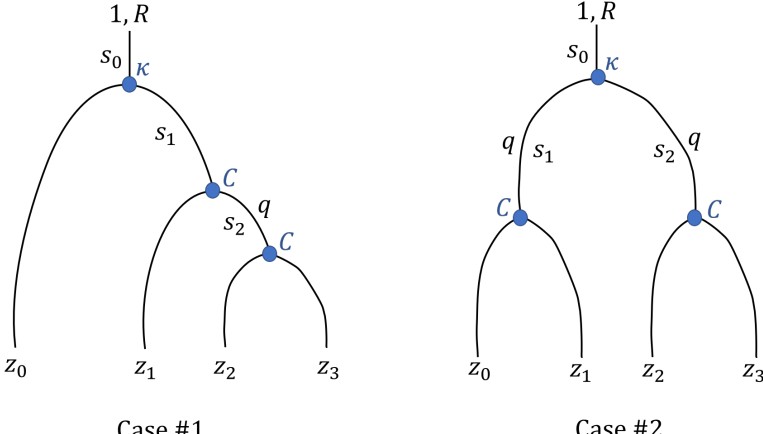

Figure 7: The two possible hierarchies for the OPE fusion trees contributing to $K_{0,4}$.

We will select one of the 3 possible channels and examine the 4-point amplitude in the *trivalent limit* where the string worldsheet has an internal line separating two 3-valent vertices. (See Fig. 6).

The internal line has a string state with $q = L_0 + \bar{L}_0 - 2 = \Delta - 2$, and in the limit where the internal particle goes nearly on-shell ($q \approx 0$) there is a contribution from string worldsheets in which the Schwinger time $s$ of the internal propagator becomes large.

Because this contribution to $A_{0,4}$ can nonlocally couple 2 distant points in target space, it can be physically distinguished from any contributions coming from small values of $s$, which behave like an approximately local 4-valent vertex. In the trivalent limit, we will freely disregard any terms in $A_{0,4}$ which can be absorbed into a 4-valent vertex.[54]

This is equivalent to saying that the dominant contribution in the trivalent limit comes from situations in which the four vertex operators are arranged with a hierarchy of scales on the sphere. Recall that when calculating $K_{0,4}$ by Tseytlin's method, we fix the Weyl frame on $S_2$. With respect to this Weyl frame, we therefore find that the points collect into groups near two of the insertions $z_0$ and $z_3$. We now project the sphere onto the plane, and use rotational symmetry to ensure that $z_3 = -z_0$. There are 2 possible cases (see Fig. 7):

1. **Singleton & triplet:** e.g. the other two insertions $z_1$ and $z_2$ are both clustered near $z_3$.

2. **Two pairs:** e.g. $z_1$ is near $z_0$, and $z_2$ is near $z_3$.

For case #1 the hierarchical assumption says that (we can number the insertions so that):

$$|z_0 - z_1| \gg |z_1 - z_2| \gg |z_2 - z_3|. \tag{69}$$

We define our Schwinger parameters as:

$$s_0 = -\log|z_0 - z_1|, \tag{70}$$

$$s_1 = \log|z_0 - z_1| - \log|z_1 - z_2| > 0, \tag{71}$$

$$s_2 = \log|z_1 - z_2| - \log|z_2 - z_3| > 0, \tag{72}$$

---

[54] In particular, this allows us to dispense with the twist term in (66) since for sufficiently long tubes the effect of integrating over twist is simply to restrict to scalars operators. Furthermore, any total derivative terms in the OPE of the form $C_{ij}^k(\partial \mathcal{O})_k$ can be absorbed into the definition of the 4-valent vertex. This means that for our purposes we can regard the indices $i$ as summing over primary scalars only. Finally, we need not worry about the question of precisely how (or whether) to try divide the regions of the moduli space with small $s$ between the 3 channels.

and at fixed positions the CFT correlator is given by

$$N_{0123}^{(\#1)} = \sum_i \kappa_{00} \, C_{1i}^0 C_{23}^i \, \exp(2s_0 - q_i s_2), \tag{73}$$

where for notational convenience we use a basis where the Zamolodchikov metric $\kappa$ is diagonal.

For case #2, the hierarchical assumption says that:

$$|z_0 - z_3| \gg |z_0 - z_1|, \tag{74}$$
$$|z_0 - z_3| \gg |z_2 - z_3|, \tag{75}$$

with the Schwinger parameters defined as:

$$s_0 = -\log|z_0 - z_3|, \tag{76}$$
$$s_1 = \log|z_0 - z_3| - \log|z_0 - z_1| > 0, \tag{77}$$
$$s_2 = \log|z_0 - z_3| - \log|z_2 - z_3| > 0, \tag{78}$$

and the CFT correlator is

$$N_{0123}^{(\#2)} = \sum_i \kappa_{ii} \, C_{02}^i C_{13}^i \, \exp(2s_0 - q_i(s_1 + s_2)). \tag{79}$$

In arranging these definitions, we have made no effort whatsoever to keep rotational symmetry on the sphere manifest. It is important that the Weyl factor on the sphere is $e^\omega = 2/(1 + z\bar{z})$, but because $q$ is nearly marginal we can get away with approximating all Weyl factors with that of the nearest point $z_0$ or $z_1$, both of which have:

$$\omega - \log(2) = \log(1 + \tfrac{1}{4} e^{-2s_0}). \tag{80}$$

Fixing a particular choice of $i$ (and hence $q$) for the internal propagator, we now integrate over the Schwinger parameters. In case #1 we have a contribution to $K_{0,4}$ that is proportional to the following integral:

$$\#1 = \int ds_0 \, ds_1 \, ds_2 \exp(2s_0 - qs_2), \tag{81}$$

$$\textbf{Range:} \quad s_1 > 0, \; s_2 > 0, \tag{82}$$

$$s_0 + s_1 + s_2 < \log(\epsilon^{-1}) - \log(1 + e^{-2s_0}/4). \tag{83}$$

The last term in (83) is the sphericity correction coming from the Weyl factor (80).[55] If we neglect this sphericity correction, we get a quadratic divergence which renormalizes the cosmological constant. This planar contribution (the "tachyon tadpole") is pure scheme and can be dropped. Instead we concentrate on the log divergence, which comes from taking the approximation:

$$\log(1 + e^{-2s_0}/4) \approx e^{-2s_0}/4. \tag{84}$$

(This is equivalent to Taylor expanding in the Ricci curvature $R$ at $z = 0$ and keeping the piece linear in $R$, i.e. the "dilaton tadpole".)

---

[55]The log(2) in (80) cancels with the fact that the vertex operator insertions are required to be $2\epsilon$ rather than $\epsilon$ apart.

Since the approximation (84) is only valid when $e^{s_0} \gg 1$, we may examine this log divergence subject to the stipulation $s_0 > 0$.[56] Applying the fundamental theorem of calculus and using the notation $E = \log \epsilon^{-1}$:

$$\#1 \;\propto\; \int_0^\infty ds_0 \, \frac{\partial}{\partial s_0} \int_{s_1, s_2 > 0}^{s_0 + s_1 + s_2 < E} ds_0 \, ds_1 \, ds_2 \, e^{-q s_2} \tag{85}$$

$$= - \int_{s_1, s_2 > 0}^{s_1 + s_2 < E} ds_1 \, ds_2 \, e^{-q s_2} = - \int_0^E ds \, (E - s) e^{-q s} \tag{86}$$

$$= \frac{1}{q^2} \left( 1 + q \log \epsilon - \epsilon^q \right), \tag{87}$$

and after applying **T1** we obtain a single (regulated) pole for the intermediate propagator in $A_{0,4}$:

$$\frac{1}{q} (1 - \epsilon^q). \tag{88}$$

Note that **T1** is equivalent to gauging out the unphysical direction $s_1$ in the LHS of (86). In the Euclidean S-matrix regime we would simply throw away the power law $\epsilon^q$ and be left with a $1/q$ divergence.

For the Lorentzian S-matrix, we would instead apply our $i\varepsilon$ prescription (43) to obtain the Feynman propagator for the internal edge:

$$\frac{-i\varepsilon}{q} \int_0^{i\infty} dE \, e^{i\varepsilon E} \left( 1 - e^{-qE} \right) \;=\; \frac{1}{q - i\varepsilon}. \tag{89}$$

If we apply **T2**, we get the same result up to terms which vanish in the $\varepsilon \to 0$ limit.

Turning our attention to case #2, we must now evaluate the integral:

$$\#2 = \int ds_0 \, ds_1 \, ds_2 \, \exp(2 s_0 - q s_1 - q s_2)), \tag{90}$$

$$\textbf{Range:} \quad s_1 > 0, \;\; s_2 > 0, \tag{91}$$

$$s_0 + s_1 < \log(\epsilon^{-1}) - \log(1 + e^{-2 s_0}/4), \tag{92}$$

$$s_0 + s_2 < \log(\epsilon^{-1}) - \log(1 + e^{-2 s_0}/4). \tag{93}$$

Following the same manipulations as in the previous case we have:

$$\#2 \;\propto\; \int_0^\infty ds_0 \, \frac{\partial}{\partial s_0} \left( \int_0^{s_0 + s_2 < E} ds_2 \, e^{-q s_2} \right)^2 \tag{94}$$

$$= \left( \int_0^{s < E} ds \, e^{-q s} \right)^2 = \frac{1}{q^2} (1 - 2\epsilon^q + \epsilon^{2q}). \tag{95}$$

If we apply **T1** we now get a term in $A_{0,4}$ which looks like a pure power law:[57]

$$\frac{2}{q} (\epsilon^q - \epsilon^{2q}). \tag{96}$$

---

[56]Neither $s_0 \lesssim 0$ nor the subleading corrections to (84) can provide the pole we are looking for, so they can be neglected in the trivalent limit.

[57]The reason why this happened, is that the fusion tree for Case #2 does not contain within it the special degeneration where $n-1$ vertex operators come together, hence unlike Case #1 there is no $\log \epsilon$ term for $\partial / \partial \log \epsilon$ to act on.

In the Euclidean S-matrix regime we could throw this term away as pure scheme. For the Lorentzian S-matrix, after applying the $i\varepsilon$ prescription we obtain:

$$\frac{-2i\varepsilon}{q}\int_0^{i\infty} dE\, e^{i\varepsilon E}\left(e^{-qE} - e^{-2qE}\right) = 2\left[\frac{1}{q - i\varepsilon} - \frac{1}{q - 2i\varepsilon}\right].\tag{97}$$

This expression, which is the $i\varepsilon$ equivalent of "pure scheme", has some peculiar properties:

On the one hand, if we evaluate (97) at exactly $q = 0$, we find that it does not vanish.

On the other hand, any *integral* of (97) with respect to a continuous test function over $q$ will necessarily vanish in the $\varepsilon \to 0$ limit, because considered as contour prescriptions it really doesn't matter whether you shift the pole by $i\varepsilon$ or $2i\varepsilon$ away from the real axis. Only the direction matters. (On the real axis, the imaginary spike coming from shifting by $2i\varepsilon$ is twice as wide, but half as tall. Hence, it limits to the same $\delta$ function.) We conclude that expressions of this nature do not make a real physical difference in the Lorentzian S-matrix.

## 5.7 Numerical coefficients at poles

The scheme dependency we discovered in case #2 leads to an important moral for interpreting Tseytlin's prescription in the S-matrix regime.

Suppose now that the sphere correlator is expanded out in the form of a $\log\epsilon$ expansion,

$$K_0 = a_0 + a_1 \log\epsilon + a_2 (\log\epsilon)^2 + a_3 (\log\epsilon)^3 + \dots ,\tag{98}$$

where, if we are sitting exactly on $n$ distinct poles, one gets $n + 1$ powers of $\log\epsilon$ in $K_0$, and hence a contribution to the $a_{n+1}$ coefficient. (For example, the 4-point correlator $K_{0,4}$ has 2 powers of $\log\epsilon$, one associated with the internal leg being on-shell, and the other being a "fake leg" associated with the special degenerations, which is on-shell for all on-shell values of the external momenta.)

Applying **T1**, we obtain the following expression for the tree-level amplitude:

$$Z_0 = a_1 + \mathbf{2}a_2 \log\epsilon + \mathbf{3}a_3 (\log\epsilon)^2 + \dots ,\tag{99}$$

where the bold faced numbers represent a symmetry factor from differentiating powers.

One might have thought that a justification of the **T1** prescription would require giving a physical explanation of why these particular numerical coefficients are correct. However, the considerations above show that this is an unrealistic ambition. Only the *integrated* size of the spike near the the poles matters physically, and this is fully determined by the behavior of the S-matrix at *generic* values of the momenta, where you get only one power of $\log\epsilon$.[58]

In this paper we do not analyze explicitly the case in which the external momenta in the Lorentzian S-matrix are taken off-shell. It should be noted however that by restricting the external legs to be (1,1), we have guaranteed that we are sitting exactly on at least one pole: namely the log divergence associated with the fake extra leg of the fusion tree. (This pole is then eliminated by **T1** or **T2**.) This pole can however be removed by going to off-shell external legs. In that case, the entire contribution of the physical on-shell S-matrix would look similar to these pure scheme dependent terms, which goes away when integrating over external momenta. Of course this is not really a problem for making physical predictions, since we are only ever supposed to integrate the S-matrix elements along the physical mass-shell.

---

[58]We have not checked whether a more naive "$1/\log\epsilon$" prescription also gives the correct integrated size of poles, as this gives rise to uglier and more ambiguous expressions.

# 6 Obtaining the classical equations of motion

In this section we take the opposite limit of finite (and real) $\epsilon$, and consider the classical (i.e. tree-level) string action $I_0$ in Euclidean signature. Recall that this was defined in 3.5 as

$$I_0 = -Z_0 = \sum_{n=0}^{\infty} A_{0,n}, \tag{100}$$

where $A_{0,n}$ is the off-shell sphere amplitude with $n$ insertions, which is obtained by applying the **T1** or **T2** prescriptions to $K_{0,n}$, the sphere partition function with $n$ insertions. Since the $n$ vertex operator insertions do not have to be marginal primaries, this is generically an off-shell perturbation to the string background CFT.

Our goal in this section is to prove that this action $I_0$ obeys the correct equations of motion, in the sense that (to all orders in perturbation theory in $n$) the equations of motion are satisfied if and only if the worldsheet theory is a CFT. (However, the explicit calculation of the action in terms of the usual target space fields will be postponed to part II.)

The basic structure of the argument in this section is as follows. First we explicitly calculate the action at orders $n = 1$ and $n = 2$ expanding around a CFT, and show that the equations of motion at this order are correct. Because we are working perturbatively in $n$, the result at these orders dominates over all higher orders *unless* the insertions are purely marginal. In fact, we will show that, without loss of generality, it suffices to consider the case of *marginal primaries* when $n \geq 3$. This case is isomorphic to the S-matrix regime, and in fact we already showed in section 5.3 (using conformal invariance) that the correct equations of motion are obtained in this case. Hence, to all orders in $n$ we obtain satisfactory equations of motion.

Although this way of constructing the proof is a bit piecemeal, the key physical idea that relates different values of $n$ is that all $\beta$ functions should be treated on an equal footing whether they come from operator dimensions (associated with the quadratic $n = 2$ action) or from nonlinear string interactions (the $n \geq 3$ part of the action).

We will assume in this section that we are perturbing around a Euclidean[59] signature CFT which is unitary (apart from the ghost sector) and has total central charge $c = 0$. (As a reminder, when we say CFT, we always mean $c = 0$ unless we indicate otherwise.) In such a unitary CFT, all operators satisfy $\Delta \geq 0$.

We will also initially take the CFT to be compact, so that the spectrum of $\Delta$ is discrete, and (as we shall see) the $n = 1$ perturbation to the action vanishes. But we will comment on the noncompact case at the end, which is a bit more difficult since normalizable perturbations cannot be exactly marginal. (An important difference in the noncompact case is that, for non-normalizable perturbations, the action might not be stationary even on a string background. An example of this is the angular $\beta$ variation of S&U in (3), where there is a nonzero first order variation of the action, i.e. $A_{1,0} \neq 0$. This will be important for obtaining a nonzero black hole entropy in part II [7])

We will take advantage of our notational convention that the index $i$ sums over *primaries* $\mathcal{P}_i$ only, in order to write the perturbation to the CFT in a way that includes an explicit sum over both primaries and non-minimally coupled terms: (cf. (23)):

$$\Delta \mathcal{L} = \sum_i \left[ \epsilon^{\Delta_i - 2} \phi^i \mathcal{P}_i + \epsilon^{\Delta_{Ri} - 2} \tilde{\Phi}^i \mathcal{P}_i R \right], \tag{101}$$

where $\Delta_{Ri} = \Delta_i + 2$ (the adjustment is due to the weight of $R$).

---

[59]We believe it is probably possible to extend these arguments directly to Lorentzian signature target space, but we leave the details to future work. It would however be extremely surprising if the right equations of motion were not also obtained in Lorentzian signature, since these equations of motion are related to the Euclidean ones by Wick rotation.

The $\tilde{\Phi}$ curvature modes do not *quite* correspond to the usual dilaton $\Phi$ of string theory. Instead, $\delta\tilde{\Phi}$ corresponds to a particular linear combination of perturbations to the dilaton $\delta\Phi$ and metric $\delta G_{\mu\nu}$, which we will work out explicitly in section 7.1.[60] These nonzero modes of $\tilde{\Phi}$ correspond to constrained modes which do not propagate in the Lorentzian signature S-matrix, because the $R$ dependence spoils Weyl-invariance even when the momentum $P_\mu$ is chosen to be null: $P^2 = 0$. However, if we consider the zero mode, a.k.a. the dilaton tadpole $\tilde{\Phi}^0$ with $P^\mu = 0$, then this *is* Weyl-invariant when integrated on the entire worldsheet (despite not corresponding to a primary in the Lagrangian).

For reasons described shortly, we also require $\Delta < \Delta_{\max} < 4$ for all terms appearing in the action (101), where the bound $\Delta_{\max}$ gets tighter at higher orders in perturbation theory. At the $n$-th order of perturbation theory, $\Delta_{\max} = 2 + 2/n$.

It is worth commenting on what is *not* included in (101). We have not bothered to write down conformal descendants of the form $L_{-1}\mathcal{O}$ or $\bar{L}_{-1}\mathcal{O}$ because they are total derivatives, and hence do not contribute to the action on a compact worldsheet.[61] These correspond to pure gauge modes in target space.

We can also exclude higher descendants like $L_{-2}\mathcal{O}$ or $\bar{L}_{-2}\mathcal{O}$ from the action because the minimum weight of a higher-descendant scalar is (2,2), i.e. $\Delta \geq 4$. The same is true for terms of the form $R^2\mathcal{P}_i$ and higher, which is good because it is not clear how to deal with them in the off-shell approach.[62]

## 6.1 Eliminating spurious tadpoles

In any compact CFT, a primary operator $\mathcal{P}_i$ with weight $\Delta > 0$ automatically has a vanishing 1 point function on the sphere. This is because, by conformal invariance, it is proportional to the vacuum 1 point function on the *plane*, which vanishes by scale invariance:

$$\langle \mathcal{P}_i(z) \rangle = 0. \tag{102}$$

One might think that this implies that $K_{0,1} = 0$. But this is false because of the possibility of what Tseytlin calls *tadpoles* in the worldsheet action, which are terms that depend only on the worldsheet metric, not on the $X$ fields. These terms are of the form:

$$t_{(p)} \frac{\epsilon^{2p-2}}{4\pi} \int d^2z \sqrt{g} R^p, \qquad p \in \mathbb{N}. \tag{103}$$

Here $t_{(0)} = T^0$ is the worldsheet cosmological constant, i.e. the zero mode of the tachyon; $t_{(1)} = \tilde{\Phi}^0$ is the Einstein-Hilbert term, and the tadpoles with $p \geq 2$ are $R^2$ and higher order tadpoles. Since we aren't sure how to deal with these higher tadpoles, we will (in the next subsection) impose a renormalizability condition which allows us to neglect them.

Each of these tadpoles has a nonzero 1 point function on a uniform sphere. The tachyon tadpole exists because the identity operator has $\Delta = 0$ and therefore the expectation value of the identity $\langle 1 \rangle$ is scale-invariant, while the tadpoles with $p \geq 1$ evade the argument above because they are not primaries; their transformation law depends on up to 2 derivatives of $\omega$.

---

[60]A linearized on-shell propagating dilaton excitation corresponds to a *different* linear combination of the dilaton $\Phi$ and the graviton $G_{\mu\nu}$, that transforms as a primary and is hence included among the $\mathcal{P}_i$'s.

[61]Technically this is only true modulo boundary terms associated with the hard disk regulator of another insertion, but it should be possible to absorb such contributions into other terms in the action. Similarly, if the dilaton terms are defined to couple to the Euler number of the *punctured* manifold, then we would need to include terms coupling to the extrinsic curvature $\int K$ at the hard disk boundaries, but this can be absorbed into a rescaling of the string fields $\phi^i$ and $\tilde{\Phi}^i$.

[62]One annoying problem is that, on a sphere of radius $r$ and curvature $R_r = 2/r^2$, terms like $(R - R_*)^2\mathcal{P}_i$ do not contribute to the 2 point function of the stress-tensor trace $\langle\langle T(0)T(z) \rangle\rangle_r$ for $z \neq 0$. Hence $T$ can vanish as an operator at radius $r$, and yet the theory is not fully Weyl invariant because it is not conformal at other radii $r' \neq r$!

Fortunately, Tseytlin's **T1** prescription eliminates the dependence of the action on the dilaton zero mode $\tilde{\Phi}^0$ tadpole, since we have:

$$Z_0(\tilde{\Phi}^0) = \frac{\partial}{\partial \log \epsilon} K_0 \propto \frac{\partial}{\partial \log \epsilon} e^{-2\tilde{\Phi}^0} = 0 \,, \tag{104}$$

leading to a flat action for the dilaton (as expected). However, since it involves differentiating by $\log \epsilon$ there is still a linear dependence of $Z_0$ on the associated beta function $\beta^R$, i.e. the renormalization of the dilaton tadpole due to other fields.[63]

Furthermore, the **T2** prescription also eliminates the first order contribution from the tachyon zero mode $T^0$. From (103)), this is just a cosmological constant $\epsilon^{-2} T^0(z)$ in the Lagrangian, so $K_0 = e^{-T_0/\epsilon^2}$ and:

$$Z_0(T_0) \propto \left( \frac{\partial}{\partial \log \epsilon} + \frac{1}{2} \frac{\partial^2}{(\partial \log \epsilon)^2} \right) e^{-T^0/\epsilon^2} \tag{105}$$

$$= \frac{4(T^0)^2}{\epsilon^4} e^{-2T^0/\epsilon^2} \,, \tag{106}$$

whose Taylor expansion in $T_0$ vanishes at $n = 0, 1$; and has a positive sign for $n = 2$ as befits a tachyon potential (since $A_0 = -I_0$).

As far as we know, nobody has proposed a prescription intended to eliminate the higher order tadpoles in the action. It is tempting to try to remove the $R^2$ pole by modifying Tseytlin's sphere prescription further, e.g. by defining

$$\mathbf{T3} := \left( 1 + \frac{1}{2} \frac{\partial}{\partial \log \epsilon} \right) \left( \frac{\partial}{\partial \log \epsilon} \right) \left( 1 - \frac{1}{2} \frac{\partial}{\partial \log \epsilon} \right) \,, \tag{107}$$

which would kill the $1$, $R$, and $R^2$ tadpoles, since these come in the action with powers of $\epsilon^{-2}$, $\epsilon^0$, and $\epsilon^2$ respectively. By adding more factors we could similarly kill an arbitrary finite number of tadpoles. These prescriptions are just as valid as **T2** from the perspective of the S-matrix arguments in section 5, but since we have not yet tested carefully their effects on all possible terms in the equations of motion (including descendants etc.) we save them for future exploration.

## 6.2 Renormalizability condition

Instead we propose to neglect the effects of these problematic tadpoles by simply not allowing terms with $R^2$ or higher couplings in our Lagrangian.

Unfortunately, such problematic terms will sometimes be introduced by renormalization even if we didn't include them originally. To keep this from happening, we need to assume a *renormalizability condition*. Recall that, if we are working at the $n$-th order in perturbation theory, we can only get a log divergence in a coupling $\phi^i$ of the form

$$\delta \phi^i \sim \log(\epsilon) \prod^n \phi^j \,, \tag{108}$$

if the dimensions satisfy

$$\dim[\phi^i] = \sum^n \dim[\phi^j] \,, \tag{109}$$

where $\dim[\phi_i] = 2 - \Delta_i$ of the corresponding operator $\mathcal{O}_i$. Otherwise one gets a power-law divergence as $\epsilon \to 0$ (if the LHS of (109) is greater than the RHS), or a power law convergence

---

[63]In fact, as we will discuss in section 6.6, there exists an RG scheme in which the contribution to $Z_0$ comes *entirely* from $\beta^R$.

as $\epsilon \to 0$ (if the RHS is greater than the LHS). In either case, we can systematically chose an RG scheme to drop these terms in a systematic way without introducing any additional dimensional scales into the RG flow.[64]

In such a scheme, in order to avoid the problematic tadpoles, it suffices[65] to perturb the CFT only with operators in the range:

$$\textbf{T1}: \quad 2 - 2/n < \Delta < 2 + 2/n, \tag{110}$$

$$\textbf{T2}: \quad 0 \le \Delta < 2 + 2/n, \tag{111}$$

where the upper end of the range prevents a log divergence of the $R^2$ tadpole which has $\Delta = 4$, and the lower end of the **T1** range prevents a log divergence in the cosmological constant, which has $\Delta = 0$.

Applied to massless fields, these restrictions still allow us to prove results about the equations of motion, *at least* to all orders in $\alpha'$. We can even, if we are careful, make some statements about the equations of motion that are nonperturbative in $\alpha'$, so long as we are working at a finite order $n$ in the conformal perturbation theory.

From the above considerations (namely the vanishing of tadpoles, together with (102)), it follows that $A_{0,1} = 0$ so long as we satisfy the appropriate renormalizability condition for our perturbations. In other words—as long as we stay within the above regimes of validity for **T1** or **T2**, *all CFTs satisfy the equations of motion*:

$$\begin{array}{ccc} \text{CFT} & \implies & \text{Solution.} \\ (\beta^a = 0) & & (E_a = 0) \end{array} \tag{112}$$

There is a sense in which this statement is necessarily valid nonperturbatively in the coupling constants, due to the vanishing of tadpoles ($n = 1$) terms as shown in the last subsection. Namely, suppose that when you perturb a $\text{CFT}_1$ by a sufficiently large value of some coupling constant $\phi$—which need not necessarily obey the renormalizability conditions above—and you end up at a new $\text{CFT}_2$. Then if you expand the action around $\text{CFT}_2$, the vanishing of tadpoles for the $\text{CFT}_2$ guarantees that it will also be a solution to **T1** or **T2**, so long as you restrict to perturbations of the $\text{CFT}_2$ which satisfy the renormalizability constraints (with the dimensions defined by the linearized beta functions near $\text{CFT}_2$).

The converse statement, that solutions to the equations of motion have vanishing $\beta$ functions, we will prove to all orders in perturbation theory in $n$ later in this section.[66]

## 6.3 Quadratic primary action

Having eliminated the linear (tadpole) piece of the action, we now confirm that at quadratic order, Tseytlin's prescription agrees with our expected result for the truncated 2-point amplitude (35). The 2-point correlator of primaries integrated on a unit sphere is fixed by conformal

---

[64]Cancellation of the convergences (terms that disappear as $\epsilon \to 0$) is equivalent to doing *nonexact* RG flow.

[65]Strictly speaking, we have only checked that these ranges are acceptable when the worldsheet metric is taken to be a uniform sphere. It is conceivable that there might be problematic changes in sign in the quadratic $n = 2$ action for other possible genus-0 metrics, although we doubt that this actually happens. In any case, as the effects of changing the Weyl frame are $O(\beta^2)$, we can use an arbitrary Weyl frame when the operators are sufficiently close to marginal.

[66]In [40] an effort was made to prove the **T1** prescription in both directions: $(\beta^i = 0) \implies (E_i = 0)$ and vice versa (on the sphere). While the justification of the former statement on a spherical worldsheet is acceptable, there is a problem with the argument given to justify that $(E_i = 0) \implies (\beta^i = 0)$. To justify it, reflection positivity was invoked to try to bound the sign of an integral of $\langle\!\langle T(z)T(0)\rangle\!\rangle$ over the sphere, but unfortunately reflection positivity does not bound the sign of the contact terms that appear when $z = 0$. See [107] for a discussion of why such contact terms make proving a c-theorem on the sphere difficult.

symmetry to be:

$$(K_{0,2})_{ij} \propto 4\pi\,\epsilon^{(\Delta_i+\Delta_j-4)} \int d^2z\,\sqrt{g}\,\big\langle \mathcal{P}_i(0)\mathcal{P}_j(z)\big\rangle_{S_2} \tag{113}$$

$$= 4\pi\,\kappa_{ij}\epsilon^{2(\Delta_i-2)} \int_{|z|>\epsilon} d^2z\, z^{-2\Delta_i}(1+z^2)^{\Delta_i-2} \tag{114}$$

$$= (8\pi^2)\kappa_{ij}\epsilon^{2\Delta_i-4}\left[\frac{1}{1-\Delta_i}(1-\epsilon^{2-2\Delta_i}) + \sum_{p\geq 0} a_p\epsilon^{(2+2p-2\Delta_i)}\right]. \tag{115}$$

Here, we have used rotational symmetry to fix one point to the origin.[67] On the second line, $\kappa_{ij}$ is the Zamolodchikov metric (which vanishes unless $\Delta_i = \Delta_j$), the first factor in the integrand is the CFT 2-point function on the plane, while the second factor is a correction due to the fact that the Weyl factor of the sphere is $e^\omega = 2/(1+z^2)$. Finally, the lower bound of the integral is the hard disk regulator.[68]

The coefficients $a_p$ are pure scheme since they can be absorbed into the order $p$-tadpole (cf. (103)), so we can drop them. Note however that Tseytlin's **T1** prescription automatically eliminates all terms proportional to the dilaton tadpole $p = 1$ (cf. (104)), while **T2** also eliminates terms proportional to the $p = 0$ tadpole, including the divergence at $\Delta = 1$ (cf. (50) and (106)).

At $\Delta = 1$, $K_{0,2} \propto \log\epsilon^{-1}$, corresponding to a log divergence of the cosmological constant. Alternatively, if we drop the $\epsilon^{2-2\Delta}$ factor (which is subleading when $\Delta < 1$), and analytically continue to all $\Delta$, we get a pole at $\Delta = 1$.[69]

We now act with the **T2** prescription.

$$A_{0,2}^{(\mathbf{T2})} = \left(1 + \frac{1}{2}\frac{\partial}{\partial\log\epsilon}\right)\left(\frac{\partial}{\partial\log\epsilon}\right)K_{0,2} \tag{116}$$

$$\propto (\Delta-1)(\Delta-2)\frac{1}{1-\Delta} \propto 2-\Delta\,, \tag{117}$$

from which we recover (35). Here we have used the fact that since $\epsilon^q = e^{(\log\epsilon)q}$,

$$\frac{\partial\,\epsilon^{2(\Delta-2)}}{\partial\log\epsilon} = 2(\Delta-2)\,\epsilon^{2(\Delta-2)}\,, \tag{118}$$

where this scaling makes sense because there are 2 operator insertions each with dimension $2-\Delta$.

If we had instead used **T1**, we would instead get

$$A_{0,2}^{(\mathbf{T1})} \propto \frac{\Delta-2}{\Delta-1}\,, \tag{119}$$

which looks similar in the vicinity of $\Delta = 2$, but fails to resolve the pole at $\Delta = 1$, because it renormalizes the cosmological constant.

Apart from the overall dimensional scaling of $\epsilon^{2\Delta-4}$ which we have not written down, our result for $A_{0,2}$ function was independent of $\epsilon$. However, this pattern will not continue to higher $n$, as for $n \geq 4$ there are poles coming from internal propagators which are not removed by Tseytlin's prescriptions **T1** or **T2**.

---

[67]To keep the expressions in this section clean, we have written the expectation value $\langle\cdot\rangle$ even though there is a proportionality constant of $\langle\!\langle 1 \rangle\!\rangle$ in $K_{0,2}$. This factor gives a generalized volume factor $V$ which we discuss in section 7.

[68]Technically the hard disk UV regulator ought to be $z > \tan\epsilon$, but this difference is pure scheme so we ignore it. Incidentally, the $z > \epsilon$ scheme has $a_1 = 1$, so in that scheme after throwing away power laws one gets a vanishing 2 point correlator for the marginal case $\Delta_i = 2$. But this difference does not affect the string action after applying **T1**.

[69]Although we are about to eliminate this particular pole, we saw in section 5.6 a similar relationship between a log divergence and the pole of the internal propagator for $n = 4$, which is not eliminated.

## 6.4 Nonminimal curvature terms

The effects of **T2** are somewhat different if we calculate the 2 point function for the nonprimary $R\mathcal{P}_i$-couplings in the action. The difference arises because while the pole in $K_{0,2}$ happens when $\Delta_i = 1$, the power of $\epsilon$ depends on the dimension of $\Delta_{Ri} = \Delta_i + 2$. Hence, we obtain (after dropping scheme dependencies):

$$K_{0,2} \propto \epsilon^{2(\Delta_{Ri}-2)} \int d^2z \sqrt{g} R^2 \left\langle \mathcal{P}(0)\mathcal{P}(z) \right\rangle_{S_2} \tag{120}$$

$$\propto \epsilon^{2\Delta_i} \left[ \frac{1}{1-\Delta_i}(1-\epsilon^{2-2\Delta_i}) \right], \tag{121}$$

where the pole at $\Delta_{Ri} = 3$ is now the result of renormalizing the bad $R^2$ term (the two $R$'s just go along for the ride and combine into $R^2$). As a result we get the structure:

$$A_{0,2}^{(\text{T2})} \propto \frac{\Delta_i(\Delta_i+1)}{1-\Delta_i}(1-\epsilon^{2-2\Delta_i}). \tag{122}$$

A few comments on this weird result are necessary. First, the zero at $\Delta_i = 0$ ($\Delta_{Ri} = 2$) makes sense because the dilaton is massless and hence the dilaton field $\Phi$ becomes marginal there.

Secondly, there is an unwanted pole at $\Delta_{Ri} = 3$ due to the renormalization of the $R^2$ tadpole, and an unwanted zero at $\Delta_{Ri} = 1$, but our assumptions (unitarity and the renormalization condition) restrict us to the range $2 \leq \Delta_{Ri} < 3$ where these don't appear. Even if we go beyond the unitarity in the Euclidean regime by allowing slightly timelike dilaton fields, we don't see any problem unless the dilaton field is highly off-shell (at the order of the string length).

Third, in the close-to-marginal range $1 < \Delta_{Ri} < 3$, the dilaton term actually has the *opposite sign* compared to a primary 2-point function of the same dimension (no matter whether we use **T1** or **T2**). This mismatch arises because $\partial/\partial \log\epsilon$ cares about the overall dimension $\Delta_{Ri}$, but the pole comes when the *primary* part of the insertion has $\Delta_i = 1$ (not at $\Delta_{Ri} = 1$). At first, one might think this is a defect in the definition of Tseytlin's sphere prescription. But in fact it is absolutely necessary! The reason is that no string action could possibly have a GR-like limit unless it replicates the *conformal mode problem* in which the Euclidean action has the wrong sign for $\sqrt{G}$ modes.

Finally, to complete our analysis of the quadratic terms, we should also examine the effects of dilaton-tachyon mixing between $R\mathcal{P}_i$ and $\mathcal{P}_i$ in the range $\Delta_i \in [0,1]$ where both fields satisfy the **T2** renormalizability condition. It suffices to examine the $2 \times 2$ coupling matrix for the $i$-th primary:

$$A_{0,2}^{(\text{T2})} \simeq \begin{pmatrix} 2-\Delta_i & -\Delta_i \\ -\Delta_i & \frac{\Delta_i(\Delta_i+1)}{1-\Delta_i} \end{pmatrix}, \tag{123}$$

and note that its determinant

$$\det A_{0,2} \simeq \frac{2\Delta_i}{1-\Delta_i}, \tag{124}$$

changes sign only at marginality ($\Delta_i = 0$) and the $R^2$-pole ($\Delta_i = 1$).[70]

## 6.5 Higher order equations of motion

Thus far we have checked the variation of the classical action $I_0$ at orders $n = 1$ and $n = 2$, and checked that it gives the correct results. We now wish to show that the action continues

---

[70]Actually, this means that the tachyon mixing resolves the spurious zero at $\Delta_i = -1$. The zero for the marginal primary at $\Delta_i = 2$ is removed but we don't vary with respect to dilatons in that range.

to behave correctly at all higher orders in $n$. Specifically, we would like to show that to all orders in perturbation theory around a CFT:

$$
\begin{array}{ccc}
\text{Solution} & \implies & \text{CFT,} \\
(E_i = 0) & & (\beta^i = 0)
\end{array}
\tag{125}
$$

which is the converse to (112). In other words, we need to check that there are no *spurious* solutions to **T1** or **T2** that are not CFTs, at least when we are perturbatively near a real CFT.

To prove this, consider any smooth curve $\mathcal{C}$ in RG space which passes through the original CFT, and whose couplings satisfy the renormalizability condition from section 6.2 at whatever order $n$ we plan to work at. We parameterize $\mathcal{C}$ by a master coupling constant $\lambda$, such that $\lambda = 0$ at the original CFT, $\lambda$ is smooth, and $\partial/\partial\lambda \neq 0$ at every point $p \in \mathcal{C}$. This allows us to control any possible small perturbation with the single parameter $\lambda$. Note that there is no requirement that $\lambda$ be a *straight line* in RG space,[71] so the $\lambda$ expansion might mix up different orders in $n$ in a $(\phi^i, \tilde{\Phi}^i)$ expansion. This is important since a hypothetical 1-parameter family of invalid solutions might not themselves lie in a straight line shot out from the original CFT. However, at $O(\lambda^n)$ in the coupling expansion, smoothness guarantees that the highest perturbation in $(\phi^i, \tilde{\Phi}^i)$ is of order $n$.

If the $\beta$ functions vanish to all orders in $\lambda$,[72] then we have a CFT to all orders $\lambda$ and we are done.

If not, then let $n-1$ be the lowest order in perturbation theory in $\lambda$ for which there is a nonzero contribution to $\beta^a_{(n-1)}$. Then we only need to calculate the equation of motion at this same order $n-1$ (which requires us to determine the action to order $n$). Everything depends on what happens at this leading order: If all the $E_a^{(n-1)}$ were to vanish, then we would have a counterexample to (125). On the other hand if (as we will show always happens) some $E_a^{(n-1)} \neq 0$, then perturbatively this will dominate all higher terms in the $\lambda$ expansion, and so there is no need to continue to higher orders.

For this reason, we may restrict attention to $n \geq 3$ (since we already did $n = 1, 2$) and to trajectories $\mathcal{C}$ which are composed of marginal perturbations only—since if the perturbation were relevant or irrelevant, we would already have a nonzero $E^{(1)}$, which would dominate over any higher order equations of motion.[73]

This means that we only need to consider perturbations with respect to the primary fields $\phi^i$ and the dilaton tadpole $\tilde{\Phi}^0$.[74] The latter simply gives us an expansion in $\lambda$ of the dilaton zero mode $\exp(-2\tilde{\Phi}^0)$, which sits outside the front of the action.

Hence, we have reduced the problem to the case where we perturb by primary marginal couplings only. But this reduces the problem to a case where we already know the answer from the S-matrix formalism! By using conformal symmetry to fix 2 points on the sphere, we showed in (61) that at "generic momentum"—which translates in this context to the statement that there are no lower order $\beta$ functions[75]—the equations of motion are proportional to beta functions:

$$
E_i^{(n-1)} = -\kappa_{ij}\beta^j_{(n-1)}.
\tag{126}
$$

---

[71]This is just as well since the concept of a straight line is dependent on the choice of RG scheme.

[72]Up to the maximum order in $n$ allowed by our renormalizability condition, which might be $\infty$ if all terms in $\mathcal{C}$ are marginal.

[73]This step in the argument uses the fact that we are in an RG scheme where marginal couplings do not lead to $\beta$ functions for relevant or irrelevant couplings. Hence, if we included any component of a non-marginal coupling in $\mathcal{C}$, at leading order in *that* perturbation, we would always get a nonzero value of the associated linear equation of motion $E_{(1)}$, which cannot be cancelled out by the marginal terms.

[74]All $\beta^{Ri}$'s besides the zero mode are irrelevant. This is because unitarity ensures that $\Delta_i > 0$ for all primaries except the identity.

[75]Since given this assumption there is only a single power of $\log \epsilon$, the value of $\beta^i$ is independent of $\epsilon$ and hence it doesn't matter if we are in the S-matrix regime or the local action regime.

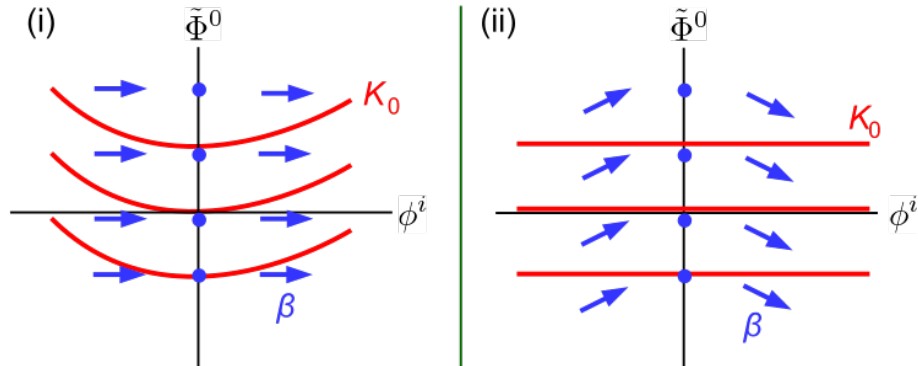

Figure 8: A 2d RG subspace of marginal couplings, including only the dilaton zero mode $\tilde{\Phi}^0$ and a single marginal primary $\phi^i$. The level sets of the sphere partition function $K_0$ are shown in red, and the vector beta function $\beta^a$ is shown in blue (we are plotting the generic case where $\beta^i$ is quadratic in $\phi^i$). Two different renormalization schemes are shown: (i) a scheme where $\beta^R = 0$; (ii) a scheme where $K_0$ is independent of $\phi^i$. These schemes are related by an RG coordinate space change: namely a shift of the $\tilde{\Phi}^0$ coordinate that is quadratic in $\phi^i$. In scheme ii, $I_0 \propto \beta^R$.

This suffices to complete the proof of (125).

## 6.6 Curci-Paffuti and dilaton schemes

It is illuminating to discuss why the possibility of non-primary couplings in the action (101) does not spoil the validity of the marginal equations of motion (126). If we calculate $I_0$ in terms of beta functions, then we have from (31) the expression (using Einstein summation):

$$I_0^{\mathbf{T1}} = \beta^a \frac{\partial K_0}{\partial \phi^a} = \beta^i \frac{\partial K_0}{\partial \phi^i} + \beta^{Ri} \frac{\partial K_0}{\partial \tilde{\Phi}^i}. \tag{127}$$

(We may as well use **T1** here, since the correction terms in **T2** coming from (32) are of minimum order $2n-2$, so they can be neglected when $n \geq 3$.)

A key point is that conformal invariance of the original CFT at $\lambda = 0$ guarantees the Curci-Paffuti property [43] that the leading order beta function is purely primary. This means that the beta function of the curvature terms vanish at leading order: $\beta_{(n-1)}^{Ri} = 0$.[76] This statement is independent of the RG scheme since we are considering a log divergence. The curvature terms can indeed appear, but only at the *next* higher order: $\beta_{(n)}^{Ri}$. This makes the dilaton equations of motion redundant with the other equations of motion, apart from the dilaton zero mode equation $\beta^R = 0$.

As a result, in calculating $I_0$ at order $n$ it turns out we will only have to worry about $\beta_{(n-1)}^i$ and the RG flow of the dilaton zero mode $\beta_{(n)}^R$, where the last term is the renormalization of the worldsheet Einstein-Hilbert term whose coefficient is $\tilde{\Phi}^0$, the dilaton zero mode. (Since the Ricci curvature $R$ has a nonzero 1 point function, it can contribute to the variation of the action at this order even though it is of one higher power of $\lambda$.) Specifically, if we vary (127) with respect to $\phi^i$, we have a primary term and a dilaton tadpole term:

$$E_i^{(n-1)} = \frac{\partial I_0^{(n)}}{\partial \phi^i} = \beta_{(n-1)}^j \frac{\partial^2 K_0}{\partial \phi^i \partial \phi^j} + \frac{\partial \beta_{(n)}^R}{\partial \phi^i} \frac{\partial K_0}{\partial \tilde{\Phi}^0}, \tag{128}$$

---

[76]This can be thought of as a Wess-Zumino consistency condition where the *coefficients* of a log divergence must still respect the symmetry that is being anomalously broken.

where we have not written out terms involving primary 1-point functions because these will vanish by (102).

The first term involves a 2 point function:

$$\frac{\partial^2 K_0}{\partial \phi^i \partial \phi^j} = 4\pi \beta_{(n-1)}^j \int d^2z \sqrt{g} \, \langle\!\langle \mathcal{P}_i(z)\mathcal{P}_j(0)\rangle\!\rangle_{\text{CFT}} \tag{129}$$

$$= 8\pi^2 \kappa_{ij}(a_1 - 1) \, \langle\!\langle 1 \rangle\!\rangle_{\text{CFT}}, \tag{130}$$

where in the last line we have used (115), but throwing away all the power laws in $\epsilon$ besides the marginal dilaton scheme dependent term $a_1$.[77] On the other hand, the dilaton tadpole is

$$\frac{\partial K_0}{\partial \tilde{\Phi}^0} \propto -4\pi \chi \, \langle\!\langle 1 \rangle\!\rangle_{\text{CFT}} = -8\pi. \tag{131}$$

These equations can only be consistent with (126) only if (as found by [43]) the two terms in (128) are proportional to each other, so that

$$\frac{\partial \beta^R_{(n)}}{\partial \phi^i} = \pi \kappa_{ij}(b + a_1)\beta_{(n-1)}^j, \tag{132}$$

with $b$ a numerical constant that could have been determined if we had been less cavalier about multiplicative constants throughout. It needs to take this form in order to add up to the scheme independent expression (126).

One confusing aspect of this story is that $a_1$ is a scheme dependent term which could be chosen to be any number, by redefining the value of $\tilde{\Phi}^0$. In particular:

- There exist RG schemes for which $a_1 = -b$ so that $E_i$ comes entirely from the *first term* in (128). In this scheme it is manifest that the equations of motion are beta functions.

- There also exist RG schemes in which $a_1 = 1$ so that $E_i$ comes entirely from the *second term* in (128), because $(K_{0,2})_{ij} = 0$. In this scheme the action $I_0$ is proportional to the dilaton tadpole $\beta^R$, making it easy to calculate.

See Fig. 8 to see how these schemes are related to each other.

The first scheme can be obtained by e.g. redefining

$$\tilde{\Phi}^0 \to \tilde{\Phi}^0 + F\phi_i\phi^i, \tag{133}$$

where $F$ is chosen so that $K_0$ is constant along surfaces of constant $\tilde{\Phi}^0$ to 2nd order in $\phi^i$. The second scheme can be obtained by instead choosing $F$ so that $\beta^R_{(n)} = 0$, where (126) guarantees that this is always possible.

For a generic momentum scattering problem, the important contribution to the S-matrix comes from a tree in which $n-1$ points come together in a log divergent way, *nested* inside of a situation where all $n$ points come together in a log divergent way. The $n-1$ divergence potentially contributes to $\beta^i$, while the $n$ divergence potentially contributes to $\beta^{Ri}$. But, it would be double counting to have the same underlying tree contribute to both $\beta$ functions simultaneously. Hence, any particular RG scheme has to *either* interpret the log divergence being due to either $\beta^i$ (inserted into the 2 point function) or $\beta^{Ri}$ (inserted into the 1 point tadpole), or perhaps some of one and some of the other.

---

[77]If the CFT partition function is not normalized to 1, there will be a factor of $e^{-2\tilde{\Phi}^0}$ out front, but since this multiplies both terms it does not affect the point we are making.

## 6.7 Noncompact CFTs

The arguments above have assumed that the CFT is compact. In a noncompact CFT there can be additional subtleties since the spectrum of $\Delta$ is now continuous. We then have to distinguish between modes that are *normalizable* with respect to the metric $\kappa_{ij}$, and those that are not.

The non-normalizable modes correspond to variations of the noncompact target space that do not fall off quickly enough at infinity [108]. Even if we expand around a CFT, these modes can have a nonzero 1 point variation $A_{0,1}$. Hence, in string theory, the background does not need to satisfy the Euler-Lagrange equations associated with such variations.[78] An example of this is the Susskind-Uglum calculation, where there is a nontrivial contribution to $I_0$ from the first order variation of the inverse temperature $\beta$ in (3).

Although we can't determine the equations of motion for non-normalizable modes without a good understanding of the boundary conditions, we still wish to argue that the equations above get us the right Euler-Lagrange equations when restricting to normalizable modes. These modes take the form of integrals over some interval of dimensions $\Delta$. This raises no particular concern for the quadratic piece of the action, as long as we keep within the range of $\Delta$'s allowed by the renormalizability condition. But it does raise some issues for the argument in section 6.5 when we restricted to marginal perturbations only, since the restriction $\Delta = 2$ is not compatible with normalizability.

Relatedly, in the noncompact case there is a major caveat with our repeated statement that it is always possible to subtract power law divergences. Consider a string scattering problem where we perturb the spacetime by some exactly marginal[79] (hence non-normalizable) deformations $\chi$ and suppose that in some RG scheme we find that $\chi$ renormalizes a family of operators $\phi^{\Delta}$ with continuous $\Delta$ in some interval $I := (\Delta_{\min}, \Delta_{\max})$ by an amount $\beta^{\Delta}(\chi)$. But if $\Delta \neq 2$ there is also a linear term in the RG equations due to the dimension, so we have:

$$\beta^{\Delta} = \beta^{\Delta}(\chi) + (2 - \Delta)\phi^{\Delta}. \tag{134}$$

Then subtracting off power law divergences is equivalent to shifting $\phi^{\Delta}$ by an amount

$$\phi^{\Delta} \to \phi^{\Delta} - \frac{\beta^{\Delta}(\chi)}{2 - \Delta}, \tag{135}$$

so as to ensure that $\beta^{\Delta} = 0$.

In the case of a compact QFT, (135) is defined whenever $\Delta \neq 2$ (the case $\Delta = 2$ corresponds to a log divergence). But in the continuous case the interval $I$ might begin at $\Delta = 2$, or pass through it, and then the shift in $\phi^{\Delta}$ will have a pole in it. The physical interpretation of this pole is that the change to the field $\phi^{\Delta}$ is non-normalizable, i.e. it does not fall off very fast at infinity.[80] (More generally, if we turn on a set of modes which are not (1,1), there will be a pole whenever (109) is satisfied.)

---

[78] At least, not without determining the appropriate boundary conditions and boundary terms (e.g. Gibbons-Hawking-like terms). It is not clear how to do this in string theory from a worldsheet perspective. In [108], a conjecture for what the boundary term of the sphere partition function was given but, to best of our knowledge, it has not been studied further or verified.

[79] By this we mean $\Delta = 2$ exactly, not that higher order $\beta$ functions vanish.

[80] Incidentally, this pole resolves a seeming paradox concerning why tree-level S-matrix amplitudes with coherent incoming and outgoing particles are nonzero, despite what we said earlier that the tree-level partition function $Z_0$ vanishes on-shell (for any CFT). The resolution seems to be that there are 2 possible pictures of the S-matrix:

- An off-shell scattering picture in which we do not shift $\phi^{\Delta}$ by the IR divergent configuration, but then the spacetime is off-shell so it is possible to have $I_0 \neq 0$ and hence a nontrivial S-matrix amplitude;

- An on-shell scattering picture in which we *do* adjust $\phi^{\Delta}$ by the IR divergent correction, but now—because the deformation is non-normalizable—we have to worry about boundary terms in the action at infinity, which need not vanish on-shell. (We don't have a good way to calculate these boundary terms from a worldsheet perspective, except to note that, since $\delta I$ vanishes for a first order perturbation to a solution, the final result for the on-shell amplitude must agree with the off-shell approach.)

Having said all of this, we still believe it is possible to show that Tseytlin's action gives the correct results in the noncompact case.

A somewhat facile argument goes as follows: at finite values of $\epsilon$, the equations of motion for $I_0$ are effectively local over some distance scale $L$ (as discussed in section 4). Hence, since there is no way for the equations of motion inside a region $\mathfrak{R}$ to "know" whether they are embedded in a compact or a noncompact geometry; hence they must be satisfied in either case. However, as it is not completely clear that every possible subregion $\mathfrak{R}$ can be embedded in an on-shell target space, this argument cannot be regarded as fully compelling. We will therefore make a more careful argument to cover the noncompact case, based on the fact that any failures of conformal invariance on the sphere QFT ought to depend smoothly on the $\beta$ functions.

To see this, let us extend our argument in section 6.5 by turning on a $\phi^i$ primary perturbation which is *normalizable*, and therefore has support on a small window of operators near $\Delta = 2$. Let us introduce a small parameter $\delta \sim |\Delta - 2|$ to keep track of the characteristic size of the deviations from marginality, and expand the action $I_0$ in a power series in $\delta$, which will take the form:

$$I_0 = b_0 + b_{1/2}\,\delta^{1/2} + b_1\,\delta + \dots \tag{136}$$

We cannot rule out the possibility of half powers of $\delta$ because in the case of massless fields, $\delta \sim \nabla^2$ where $\nabla^2$ stands for a second order Laplacian.[81] The expansion is a valid one because we can take $\delta$ to be arbitrarily close to 0 while still having the modes be normalizable.

Now all effects of the $O(\delta^0)$ term, because they are independent of $\delta$, can be calculated from the marginal case ($\delta = 0$) and hence can be treated as if they were a purely marginal perturbation for the purposes of section 6.5. Similarly, the $O(\delta^{1/2})$ term—if it exists—is also effectively marginal, since the first order beta function $\beta^{(1)} \propto \delta$. So for these terms we have, just as in the marginal case:

$$E_i^{(n-1)} = -\kappa_{ij}\beta_{(n-1)}^j, \tag{137}$$

which as a reminder we derived in section 5.3 using the conformal invariance of marginal vertex operators on the worldsheet sphere.

For the remaining $O(\delta^1)$ and higher terms, we use the principle that *all failures of conformal invariance are proportional to beta functions* to write:

$$E_i^{(n-1)} = -\kappa_{ij}\beta_{(n-1)}^j + O(\delta)\beta_{(k)}^j X_{ij}, \tag{138}$$

where $X_{ij}$ represents whatever corrections to conformal invariance arise due to the perturbation $\phi^i$ not being perfectly marginal. Now if $k < n-1$, by induction we know that $\beta_{(k)}^j$ is proportional to a (normalizable) lower order equation of motion $E_i^{(k)}$. As discussed in section 3.3 such terms can always be compensated for by making a local redefinition of fields, so let us assume this has been done. If $k = n-1$, then since $\kappa_{ij}$ is nondegenerate, we have that under an arbitrarily small perturbation

$$\kappa_{ij} \to \kappa_{ij} + O(\delta)X_{ij}, \tag{139}$$

it remains nondegenerate for sufficiently small $\delta$. We need not consider $k > n-1$ because it is subleading in $n$. Hence, $E_i^{(n-1)} \neq 0 \iff \beta_{(n-1)}^i \neq 0$ for primary perturbations.

For completeness we also need to consider cases involving the curvature modes $\tilde{\Phi}^i$. Using our renormalization condition, Curci-Paffuti, and the fact that curvature modes can't affect the

---

This is analogous to the on-shell vs. off-shell methods for computing black hole entropy, which we will discuss further in part II [7].

[81]E.g. when expanding around a stable, translation-invariant, but nonisotropic background, any effect which depends linearly on some component of the momentum $P^\mu$ of some massless particle will show up at half order in a $\delta \sim P^2$ expansion.

beta functions of primaries, the only additional type of beta function we need to consider is if $\tilde{\Phi}^i$ is renormalized by a single $\tilde{\Phi}^j$ insertion multiplied by some order $(n-2)$ of the primaries $\phi$. Let us call this beta function $\beta^{Ri}_{(n-2,1)}$ to keep the orders in the primaries separate from the orders in the curvature terms.[82] Since this case is not conformally invariant,[83] we directly plug into (31) to obtain for the dilatonic equation of motion:

$$E^{(n-2,1)}_{Ri} = \frac{\partial I^{(n-2,2)}_0}{\partial \tilde{\Phi}^i} = \beta^{Rj}_{(n-2,1)} \frac{\partial^2 K_{0,2}}{\partial \tilde{\Phi}^i \partial \tilde{\Phi}^j} \tag{140}$$

$$\propto 4\pi \beta^{Rj}_{(n-2,1)} \int d^2 z \sqrt{g} R^2 \langle \mathcal{P}_i(0) \mathcal{P}_j(z) \rangle. \tag{141}$$

But this 2 point function is convergent because $\Delta_i \approx 0$, and is approximately proportional to $\kappa_{ij}\beta^{Rj}$ up to $O(\delta)$ corrections. (There is no dilaton tadpole $K_{0,1}$ term in this equation since $\beta^R_{(n-2,2)} = 0$.) We therefore have a non-degenerate expression and hence $E_{Ri}^{(n-2,1)}$ is nonzero whenever there is a nonzero $\beta^{Rj}_{(n-2,1)}$. This completes the proof that the equations of motion are satisfied in the noncompact case.

## 6.8 Comments on supersymmetry

Since all our arguments so far have concerned bosonic string theory, we quickly describe how we expect things to be different in superstring theory, without doing a careful analysis. For specificity we consider the RNS formalism of type II strings, although everything we say should generalize naturally to the heterotic case with suitable adjustments.

Obviously, we will now need to gauge fix the super-ghost sector $\beta$ and $\gamma$ on the worldsheet. The easiest case to consider is when all of the insertions on the worldsheet are NS-NS (but not necessarily marginal), so that they preserve global supersymmetry.[84] In this case, we expect that an analogue of our off-shell gauge-fixing procedure from section 3.2 will leave us with a zero mode sector equal to the *superconformal* Killing group SCKG on the sphere. According to Tseytlin [9,64], the volume of the gauge orbits of this supergroup go like

$$\mathrm{Vol}(\Omega) \sim \log \epsilon + O(1), \tag{142}$$

without any leading order $1/\epsilon^2$ divergence, assuming that our regulator respects supersymmetry (which the hard disk certainly does not!) This is because supersymmetry prevents divergences in the worldsheet cosmological constant. As a result, it is now possible to use **T1** without ever worrying about the **T2** correction.

As partial confirmation of this, we note that in a unitary super-CFT, deformations of the Lagrangian that preserve SUSY take the form:[85]

$$\mathcal{S} \sim G_{-1/2} \bar{G}_{-1/2} \mathcal{O}. \tag{143}$$

---

[82]It does not matter if the trajectory $\mathcal{C}$ through RG space mixes the 0th and 1st orders in $\tilde{\Phi}$, as we use distinct equations of motion in each case.

[83]There is, however, another approximate symmetry, whereby unitarity tells us that any CFT $n$-point correlator is independent of the position of the $R\mathcal{P}_i$ insertion in the the $\Delta_i \to 0$ limit. This is because $\langle \nabla^2 \mathcal{P} | \nabla^2 \mathcal{P} \rangle = \langle \mathcal{P}|L_1 \bar{L}_1 L_{-1} \bar{L}_{-1}|\mathcal{P} \rangle = 4\langle \mathcal{P}|L_0 \bar{L}_0|\mathcal{P} \rangle = O(\delta^2)$, and the only mode on a compact worldsheet which is annihilated by $\nabla^2$ is the zero mode. By reflection positivity, this implies that all divergences which depend on nonzero modes of a nearly marginal $R\mathcal{P}_i$ insertion must be $O(\delta)$ or smaller. But divergences are local and thus cannot be independent of the position of the $R\mathcal{P}_i$ insertion! Hence they are $O(\delta)$, and we can neglect them by same argument as in the pure primary case. This provides an alternative argument to the one in the main text.

[84]Or at least, that global SUSY *would* be preserved if the theory were on the plane. It is not possible for a unitary, *nonconformal* QFT to preserve global SUSY on the sphere, because any $\{Q, Q^\dagger\}$ gives us a positive Hamiltonian, but there are no everywhere-timelike Killing fields in de Sitter.

[85]This is because in the super-conformal algebra, $\{G_{-1/2}, G_{-1/2}\} = 2L_{-1}$, and hence a further application of the SUSY generator $G_{-1/2}$ will always produce a total derivative term, which vanishes when integrated on the worldsheet.

Now from unitarity, $\Delta_{\mathcal{O}} > 0$ and hence $\Delta_{\mathcal{S}} > 1$. So even in an unfavorable situation where the GSO projection fails to remove all of the tachyons[86] from the spectrum, they are still always above the cosmological constant pole at $\Delta = 1$ in the 2-point function (115). Since we are not using the vertex operators to gauge-fix the superconformal zero modes, we will use (143) for all $n$ vertex operators, and not for $n-2$ as is usually done.

Since the R-R fields break SUSY by introducing a twist, it is probably necessary to treat them separately. There are always an even number of R-R insertions on the worldsheet, and a single pair suffices to break all of the supersymmetry zero modes. We therefore suspect that the easiest way to put R-R insertions off-shell is to first use them to fully fix the supersymmetry (leaving only the bosonic CKG group unfixed) and then integrate over all $n$ positions as one does in the off-shell bosonic string theory. Since R-R operators always have $\Delta \geq 2$ in a unitary theory, there does not seem to be any obvious reason why the application of **T1** should fail in their case either.

# 7 C-functions and actions

## 7.1 Fields in the nonlinear sigma model

Since the previous discussion has been stated abstractly in terms of operators of dimension $\Delta$, it is worth showing explicitly what the result is for a NLSM defined in terms of the usual graviton $G^{\mu\nu}$ and dilaton $\Phi$ fields.

As we will show by explicit calculation in part II [7], at leading order in $\alpha'$, the QFT sphere partition function $K_0$ of a NLSM takes the form

$$K_0 = \frac{1}{g_s^2} \int \mathrm{d}^D X \sqrt{G}\, e^{-2\Phi}(1 + O(\alpha')). \tag{144}$$

In fact, to arbitrary orders in $\alpha'$ (and at fixed $\epsilon$) it is always possible to adopt an RG scheme [47, 93, 94] in which the dilaton $\Phi$ is shifted by a local counterterm so that

$$K_0 = V := \frac{1}{g_s^2} \int \mathrm{d}^D X \sqrt{G}\, e^{-2\Phi}, \tag{145}$$

exactly.[87] Here $V$ is a *generalized* volume because it is weighted by the factor $e^{-2\Phi}$, coming from the dilaton coupling to the Euler number $\chi = 2$. In fact, if we restrict attention to RG schemes in which $G_{\mu\nu}$ and $\Phi$ transform as tensors, then this is (up to a change in $g_s$) the *unique* positive covariant ultralocal integral that scales like $\exp(2\Phi^0)$ under a shift of just the dilaton zero mode $\Phi^0$. In particular, there is no covariant way to remove the dependence on the metric via $\sqrt{G}$.

We can, of course, change the coupling constant $g_s$ by shifting the dilaton $\Phi$ by a constant. However, this will also affect the value of various on-shell scattering processes. So another way to put this is that the overall multiplicative factor in front of the leading term in $K_0$ (and hence in front of $I_0$[88]) is fully determined by the on-shell data, via the requirement that we use the same CFT on all worldsheets regardless of topology.[89]

---

[86]This class of tachyons should not be confused with *the* bosonic tachyon, in particular they have no associated tadpole in $K_{0,1}$.

[87]This is a generalization of one of the RG schemes defined in section 6.6, but taken beyond quadratic order. Algebraically it is easy to define the Tseytlin scheme nonperturbatively in $\alpha'$, but there is no guarantee that the field redefinition is local in target space unless we stop at a finite order in the derivative expansion.

[88]However, we warn the reader that calculating the numerical factor in front of this multiplicative constant would require keeping track of several measure and kinematic factors which we have dropped by the wayside.

[89]In particular the value of $K_0$ on-shell is independent of the sphere radius $r$ because we require $c = 0$.

If we now vary the partition function (145), we obtain:

$$\delta K_0 = -\frac{2}{g_s^2} \int \mathrm{d}^D X \sqrt{G}\, e^{-2\Phi} \left( \delta\Phi - \frac{1}{4} G^{\mu\nu} \delta G_{\mu\nu} \right).$$

(146)

It follows from (146) that the sphere 1-point function $K_{0,1}$ is nonvanishing, not just for the dilaton, but *also* for certain conformal variations of the metric $G_{\mu\nu}$. This requires that the CFT operator $\mathcal{O}_{\sqrt{G}}$ associated with varying the conformal factor has an anomalous dependence on the worldsheet curvature $R$, so that its expectation value $\langle \mathcal{O}_{\sqrt{G}} \rangle$ is different on the sphere and the plane,[90] because all marginal 1-point functions vanish on the plane.

It is this combination of variations in (146) that Tseytlin refers to as the *dilaton tadpole*, which is the variation of the $\tilde{\Phi}^0$ mode which we referred to extensively in 6.

Let us now see how the target space fields break up into the primary and curvature terms in the worldsheet Lagrangian (101), which we used in conformal perturbation theory. (There we omitted the pure gauge modes, but below we will include them.) For this task, we will also need to identify the *nonzero* modes $\tilde{\Phi}^i$ that are coefficients of the curvature terms in (101). Given (145), an obvious candidate (in position space) is the logarithm of the generalized volume element:

$$\hat{\Phi} = \Phi - \frac{1}{4} \log \det G,$$

(147)

yet there is a possible ambiguity due to the addition of a total derivative term to the generalized volume integrand (145), in which case $\tilde{\Phi}$ and $\hat{\Phi}$ might differ.

For simplicity, we now restrict attention to a flat Euclidean background $G_{\mu\nu} = \delta_{\mu\nu}$ with zero dilaton $\Phi = 0$, plus a *first order* perturbation to the metric $\delta G_{\mu\nu}$ and dilaton $\delta\Phi$. The pure gauge modes (corresponding to $L_{-1}$ and $\overline{L}_{-1}$ descendants) must be diffeomorphisms, hence in momentum space they take the form:

$$\text{pure gauge:} \qquad \delta G_{\mu\nu} = P_\mu \xi_\nu + P_\nu \xi_\mu,$$

(148)

$$\delta\Phi \quad = 0.$$

(149)

It may be observed that this gauge transformation can affect the value of $\hat{\Phi}$ as defined in (147); as this is not a curvature mode in the worldsheet Lagrangian, it follows that $\tilde{\Phi} = \hat{\Phi}$, only if we impose the gauge $P^\mu P^\nu G_{\mu\nu} = 0$ on the gravitons; otherwise there will be a correction which depends nonlocally on $\delta G_{\mu\nu}$.

Meanwhile the primary modes are defined by the requirements that (i) they are orthogonal to the total derivatives in the 2pt function, and (ii) they do not transform anomalously under a $\nabla^2 \omega$ Weyl rescaling on the worldsheet. This gives the modes:

$$\text{primary:} \qquad \delta G_{\mu\nu} = h_{\mu\nu} \ \ (\text{with } P^\mu h_{\mu\nu} = 0),$$

(150)

$$\delta\Phi \quad = h^\mu_\mu / 4,$$

(151)

where the transverse condition on $h_{\mu\nu}$ comes from (i), and (ii) can be verified by checking that the variation does not contribute to (147)—this suffices because the anomaly in the metric operator $: (\partial_A X^\mu)(\partial^A X^\nu) e^{iP\cdot X} :$ is the same for each $P_\mu$ mode, and thus $\delta\Phi$ should take the same form for a nonzero mode sector as it does for the zero mode sector. (Note that the usual "dilaton primary", because it is a primary, is contained in the scalar modes of $h_{\mu\nu}$ and does not involve a nonzero value of $\delta\tilde{\Phi}$ at all!) As we are in Euclidean signature, these modes are all off-shell when $P_\mu \neq 0$.

---

[90]The precise reason for this is somewhat dependent on your choice of regulator scheme, but when using a target-space covariant heat kernel method (which we use in part II) it arises as a combination from the heat kernel regulation of $: \partial_A X^\mu \partial^A X_\mu :$ and the measure factor.

Finally, a shift in the 2d curvature modes $\tilde{\Phi}$ has no effect on the Lagrangian of a locally flat worldsheet ($R = 0$), and is therefore defined by shifting the $\Phi$ field alone:

$$\text{curvature:} \qquad \delta G_{\mu\nu} = 0\,, \tag{152}$$

$$\delta \Phi \quad = \delta \tilde{\Phi}\,. \tag{153}$$

Allowing all the modes together, we therefore have:

$$\delta G_{\mu\nu} = h_{\mu\nu} + 2P_{(\mu}\xi_{\nu)}\,, \tag{154}$$

$$\delta \Phi \quad = \delta \tilde{\Phi} + h^{\mu}_{\mu}/4\,. \tag{155}$$

This may be inverted to obtain the coefficients used for conformal perturbation theory, but in order to obtain simple expressions that appear local in position space, we will assume transverse gauge ($\xi_{\mu} = 0$). Then:

$$\delta h_{\mu\nu} = \delta G_{\mu\nu}\,, \tag{156}$$

$$\delta \tilde{\Phi} \quad = \delta \Phi - \tfrac{1}{4}\delta G_{\mu\nu}\,, \tag{157}$$

where the last expression verifies that $\tilde{\Phi} = \hat{\Phi}$ in this gauge.

As a consistency check, it can be seen that

$$\left.\frac{\delta K_0}{\delta h_{\mu\nu}}\right|_{\tilde{\Phi},\xi} = 0\,, \tag{158}$$

because $\langle\!\langle P_i \rangle\!\rangle = 0$ for massless primaries in a CFT. Furthermore, even though $\Phi \neq \tilde{\Phi}$ in general, their basis vectors do agree in the two coordinate systems:

$$\left.\frac{\delta}{\delta\tilde{\Phi}}\right|_{h,\xi} = \left.\frac{\delta}{\delta\Phi}\right|_{G}\,. \tag{159}$$

Hence, a shift in the zero mode $\tilde{\Phi}^0$ in conformal perturbation theory, is equivalent to a shift in the zero mode $\Phi^0$ in the usual string fields.

## 7.2 Central charge action

In the Tseytlin scheme (as defined in 6.6), the action comes solely from the variation of the $\tilde{\Phi}$ dilaton tadpole:

$$I_0^{\mathbf{T1}} = -\frac{\partial K_0}{\partial \log \epsilon} = 2K_0 \beta^R \tag{160}$$

$$= \frac{2}{g_s^2} \int \mathrm{d}^D X \sqrt{G}\, e^{-2\Phi} \tilde{\beta}^{\Phi} \tag{161}$$

$$= \frac{2}{g_s^2} \int \mathrm{d}^D X \sqrt{G}\, e^{-2\Phi} \left( \beta^{\Phi} - \frac{1}{4} G^{\mu\nu} \beta^G_{\mu\nu} \right)\,, \tag{162}$$

where $\tilde{\beta}^{\Phi} := \beta^{\tilde{\Phi}}$ is the position space expansion of the curvature-dependent beta functions $\beta^{Ri}$, weighted by the generalized volume element $\sqrt{G}e^{-2\Phi}$.

On a weakly curved off-shell background, these beta functions can be calculated in the NLSM (e.g. [75]) and are (at leading order in $\alpha'$):[91]

$$\beta_{\mu\nu}^{(G)} = \alpha' R_{\mu\nu} + 2\alpha' \nabla_\mu \nabla_\nu \Phi, \tag{165}$$

$$\beta^{(\Phi)} = -\frac{1}{2}\alpha' \nabla^2 \Phi + \alpha' (\nabla\Phi)^2. \tag{166}$$

Eq. (161) is called the *central charge action*, because in a CFT with central charge $c$, the Curci-Pafutti theorem tells us that $\tilde{\beta}^\Phi$ is constant in target space and proportional to the central charge $c$ (which of course equals 0 on an on-shell string background). Using this observation, Tseytlin was able to leverage his action into a perturbative argument for the c-theorem for the NLSM [46, 47], which we review in what follows.

It is not immediately obvious how to get a monotonic flow given that the 2-point amplitude $A_{0,2}$ for the curvature mode $\tilde{\Phi}$ has the opposite sign from the primary perturbations of the same dimension $\Delta$, as shown in section 6.4. However, it turns out that it has definite signature if we *first* constrain $\tilde{\Phi}$ to solve the $E_{\tilde{\Phi}} = 0$ equations, as we will explain in more detail in section 7.4.

This doesn't quite get us the usual c-theorem, because the constraint $E_{\tilde{\Phi}} = 0$ is a little bit too strong: it requires that $c = 0$, so e.g. it doesn't allow us to consider RG flows between two different CFTs (since at least one will have central charge $c \neq 0$.) To deal with this problem we need to make a small modification which we describe in the next section.

## 7.3 Trace formula for T1

We start by describing the relationship of **T1** to the trace $T$ of the stress-energy tensor.

Since $\epsilon$ is the only dimensionful coupling constant in the QFT, on a sphere a factor of $\log \epsilon$ always takes the form $\log(\epsilon/r)$ to keep the argument dimensionless. Hence, the **T1** prescription is equivalent to differentiating with respect to the log radius of the sphere, i.e. inserting the trace of the stress-tensor into the sphere partition function:

$$I_0^{\mathbf{T1}} = \frac{\partial}{\partial \log r} K_0 = 4\pi \langle\!\langle T \rangle\!\rangle_{\text{QFT}}, \tag{167}$$

where we may use rotational symmetry to place the trace at $z = 0$. This formulation of the action makes it clear that, for a CFT with a trace anomaly, the **T1** action will be proportional to the central charge $c$, for example $D - 26$ in bosonic string theory. This means in particular that a CFT with $c \neq 0$ violates the dilaton equation of motion.

More generally, the **T1** formula may be thought of as a method to determine the log divergence associated with the Euler number $\chi$ on the worldsheet. As we saw in section 5.6, this is the part of $K_0$ that gives the physically relevant contribution to tree-level scattering amplitudes. However, **T1** fails when there is a cosmological constant since it can't tell the difference between a trace $T$ which is due to curvature $R$, and a $T$ which is due to vacuum energy. So in such situations we will need to use **T2** instead.

---

[91]Ref. [75] also showed that, in string field theory, the $n = 1$ tadpoles emitted by the nonvanishing $\beta$ function leads to a Fischler-Susskind shift [95, 96] in the background fields, which is at leading order in $\log \epsilon$ given by:

$$\delta G_{\mu\nu} = \beta_{\mu\nu}^{(G)} \log \epsilon, \tag{163}$$

$$\delta \Phi = \beta^{(\Phi)} \log \epsilon. \tag{164}$$

## 7.4 c-theorem for the nonlinear sigma model

To find an object which is stationary even for CFTs with $c \neq 0$, we can instead consider the *expectation value* of the trace:

$$C = \langle T \rangle = \frac{\langle\langle T \rangle\rangle}{\langle\langle 1 \rangle\rangle} = \frac{I_0}{V} \tag{168}$$

$$= \frac{\int d^D X \sqrt{G}\, e^{-2\Phi}\, \tilde{\beta}^{\Phi}}{\int d^D X \sqrt{G}\, e^{-2\Phi}}\,, \tag{169}$$

which is of course equal to $c$ for a CFT. The difference is that $I_0$ is an *integral* over target space, while $C$ is a (weighted) *average* over target space.[92]

The variation of the sphere action can now be written as:

$$\delta I_0 = \delta(CV) = V\,\delta C + C\,\delta V\,. \tag{170}$$

From this it can be seen that, in order to be able to convert between the two types of stationarity

$$\delta I_0 = 0 \iff \delta C = 0\,, \tag{171}$$

we will need to have either $C = I_0 = 0$ (the conditions for a string background) or else restrict attention to variations with $\delta V = 0$.[93]

Having constrained the generalized volume $V$, a theorem of Oliynyk, Suneeta, & Woolgar [109] shows that if we consider $C[\tilde{\Phi}]$ (168) as a function of the modes $\tilde{\Phi}^i$, there exists a unique maximum, at least at leading order in $\alpha'$. (See [110] for the noncompact case.) Tseytlin argued, based on experience of subsequent orders in $\alpha'$, that this property would continue to be true to all orders in perturbation of $\alpha'$, as long as we stay within the validity of the perturbative expansion [47]. A clear explanation for why this must be the case, based on the nature of perturbation theory, was recently provided by [111].

At this point we may construct a perturbative c-theorem for the NLSM as follows: Let $\{\phi^i\}$ be a set of primary couplings without specifying $\tilde{\Phi}^i$ (as the latter cannot be determined by measuring flat space correlations). We now perform the following steps:

(i) *Solve* for $\tilde{\Phi}^i$ using its own equation of motion, by maximizing $C[\tilde{\Phi}]$ at fixed values of the primaries $\phi^i$.[94] This gives us a solution to the $\tilde{\Phi}^i$ equation of motion, for which

$$\frac{\partial C}{\partial \tilde{\Phi}^i} = 0\,. \tag{173}$$

This holds even for the zero mode $\tilde{\Phi}^0$, as the dependence on $\tilde{\Phi}^0$ cancels between the numerator and denominator of (169).

---

[92]Here we are implicitly assuming compactness so both are defined. In the noncompact case, usually $V = \infty$, so at most one of the expressions will be defined. For normalizable off-shell perturbations to a $c = 0$ string background, only the integral is well defined, while for a CFT with central charge $c$ only the average is well defined.

[93]The latter condition may also be implemented by adding a Lagrange multiplier $C'$ to the action $I$ and dividing by $V_0$ to obtain [47]:

$$C = \frac{1}{V_0}\left[I_0 - C'(V - V_0)\right]\,, \tag{172}$$

which constrains the generalized volume to $V = V_0$ and sets $C' = -C$. It does not matter that the RG flow does not preserve the condition $V = V_0$ since we only need to impose that condition at one particular RG scale $\epsilon$.

[94]This maximization leaves undetermined an arbitrary shift in its zero mode $\tilde{\Phi} \to \tilde{\Phi} + a$ which does not affect what follows.

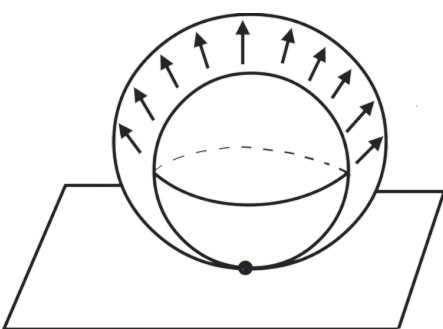

Figure 9: **T2** is equivalent to differentiating $\langle\langle T(0)\rangle\rangle$ with respect to a stereographic resizing of the sphere that leaves unchanged the tangent plane to the origin—shown as the black dot at the south pole of the sphere. This Weyl transformation is the combination of a uniform Weyl rescaling and a conformal isometry of the sphere. It may be implemented with an integral over a second insertion of $T$, weighted by $1 - \cos\theta$, where $\theta$ is the angle to the origin.

(ii) *Evaluate C* on the resulting solution $\{\phi^i, \tilde{\Phi}^i\}$ to obtain a function $C(\phi^i)$ over $\{\phi^i\}$ alone. The RG flow of this quantity is now given by (from section 6):[95]

$$\frac{\delta I_0}{\delta \phi^i} = -\kappa_{ij}\beta^j. \tag{174}$$

This is true even though there may be an RG flow $\beta^{Ri}$ of the curvature terms, because of stationarity w.r.t. $\tilde{\Phi}^i$ (173). Hence under RG flow:

$$\frac{dI_0}{dt} = \frac{\partial \phi^i}{\partial t}\frac{\partial I_0}{\partial \phi^i} = -\kappa_{ij}\beta^i\beta^j \leq 0, \tag{175}$$

where the last inequality becomes strict when $\beta^i \neq 0$.

This tells us that $I_0$ decreases monotonically under any RG flow perturbatively close to an on-shell string background. Hence, this $C$-function is monotonically decreasing along the $\beta^i$ trajectories. It is also stationary at fixed points, where $C = c$.

This C-function is numerically different from the one defined by Zamolodchikov [17]; in particular it has better IR behavior due to the compactification of the worldsheet into a sphere. This is why it is approximately local in target space, and defines an irreversible flow even for noncompact CFTs. We will compare these in more detail in [112].

## 7.5 Trace formula for T2

Just as it is possible to write **T1** as a 1-point function of the trace (167), it is similarly possible to write the **T2** prescription as a 2-pt function of the trace:

$$I_0^{\mathbf{T2}} = \left(\frac{\partial}{\partial \log r} - \frac{1}{2}\frac{\partial^2}{\partial (\log r)^2}\right)K_0 \tag{176}$$

$$= -\int d^2z \sqrt{g}\, \frac{|z|^2}{1+|z|^2}\, \langle\langle T(0)T(z)\rangle\rangle_{\text{QFT}}, \tag{177}$$

---

[95]As discussed in 6.4, this formula only needs to be valid at leading nonvanishing order $n$ in perturbation theory. Also, we have absorbed some unimportant positive numerical coefficients into $\kappa_{ij}$, including a power of $\epsilon$ in the non-marginal case.

where $|z|^2/(1+|z|^2) = \frac{1}{2}(1-\cos\theta)$.[96] But in any correlator, the trace of the stress-tensor is equivalent to a Weyl transformation:

$$T(z) = \frac{\delta}{\delta\omega}. \tag{178}$$

Hence, on a sphere of general radius $r$, the integrated $T(z)$ factor in (177), which can also be written as:

$$r^4 \int d\theta\, d\varphi\, \sqrt{g}\,(1-\cos(\theta))\, T(\theta,\varphi), \tag{179}$$

generates a stereographic map between two spheres of slightly different radii $r$, but *without* rescaling the tangent plane at $z = 0$, as shown in Fig. 9. In this respect it differs from a uniform rescaling of the sphere generated by a constant trace, because the uniform scaling would also rescale the other stress tensor $T(0)$ as a weight 2 object, while the former one leaves it alone. This accounts for the linear $\partial/\partial(\log r)$ term in the **T2** prescription.

In particular, this means that the contribution of the cosmological constant to $T(0)$ will cancel out of the expression, as we found in section 6.1. The expression (177) is therefore roughly equivalent to differentiating the stress-tensor $T(0)$ with respect to the curvature tensor $R$. So long as there are no $R^2$ and higher effects to worry about, this again picks out a log divergence associated with the Euler number $\chi$.

## 7.6 Relation to planar c-theorem

We now describe the close relationship between **T2** and the planar c-theorem.

Although it is not usually presented in this way, Zamolodchikov's planar C-function is equivalent to the following integral over a disk of radius $r_*$:[97]

$$C \propto -\int_{|z| < r_*} \mathrm{d}^2 z\, |z|^2 \left\langle T(z) T(0) \right\rangle_{\mathrm{QFT}}. \tag{180}$$

Note that the factor of $|z|^2$ guarantees that the formula is dimensionless except for its dependence on $r_*$. As we plan to detail in forthcoming work [112], this formula allows for an elegant proof of all aspects of the planar $c$-theorem:

1. In a unitary CFT, reflection positivity of $\langle T(z)T(0) \rangle$ for $z \neq 0$ guarantees that this expression is monotonically decreasing with increasing $r_*$.

2. In a CFT with a central charge $c$, $C = c$, because by (178), the integrated stress-tensor factor $\int \mathrm{d}^2 z |z|^2 T(z)$ generates a Weyl transformation that shifts the curvature $R$ of the spacetime metric inside the disk; then $T(0)$ measures the resulting trace anomaly. This is because $\beta^R$ is proportional to $c$ in a CFT.

3. $C$ is stationary to first order around any such CFT. This is because $T \sim \beta$ and there are two $T$'s so the action is order $O(\beta^2)$. Contact terms with like $\beta^{Ri}\langle \mathcal{P}_i\rangle_{\mathrm{QFT}}$ (with $\Delta_i > 0$) do not provide a loophole here because such 1 point functions vanish in any compact CFT, and so are themselves proportional to $\beta$ functions.

---

[96]In evaluating the 2 point function of $T$, it is important to note that a hard disk regulator does not necessarily prevent the two stress-tensors from approaching close to each other. Rather $T$ is proportional to $\delta/\delta\omega$ acting on a partition function in which the *vertex operators* have hard disks around them.

[97]In the literature this formula usually appears as a "sum rule" and there is a UV cutoff preventing contact terms from the two traces touching each other. But in our application the contact term is desirable, as it provides the central charge of the CFT!

4. Conversely, $C$ is not stationary at a non-CFT, because it monotonically decreases along the RG flow itself.

We now comment on the differences between the planar central charge $C$ and **T2**. For **T2**, the sphere radius plays the same scaling role that the disk radius does for $C$, but because the sphere is finite, it does a better job of cutting off IR divergences. Furthermore, just like the case of **T1**, the fact that we have a sphere ($\chi = 2$) gives us the important $\exp(-2\Phi)$ dilaton prefactor in the string action, variations with respect to which enforce the $c = 0$ constraint of string backgrounds.

Since (177) uses the unnormalized amplitude $\langle\!\langle \cdot \rangle\!\rangle$ rather than the expectation value $\langle \cdot \rangle$ it is once again, like the $C$ of section 7.4, an integral over target space rather than an average.

We do not know how to make a fully general (nonperturbative) proof of the monotonicity results 1 and 4 on the sphere, except in some special cases [112]. On the other hand, the proofs of 2 and 3 carry over immediately to the classical string action, at least if the target space is compact. This once again gives us from another perspective the result (112), that all CFTs are stationary.

# 8 Discussion

In this paper, we reviewed and extended Tseytlin's off-shell NLSM formalism. This is a first quantized formalism, in which one takes the worldsheet field theory to be a non-conformally invariant QFT. (In our work we do not need to assume that this QFT takes the form of a standard NLSM; so we can also consider highly non-geometrical string compactifications.)

The first goal of this paper was to convince you that—contrary to beliefs of many in the string theory community—there is nothing inherently ill-defined about off-shell string theory (at least perturbatively). Specifically, we argued in section 3.1 that ambiguities coming from specifying the Weyl factor $\omega$ on the worldsheet, can always be fully absorbed by a corresponding field redefinition ambiguity in the target space fields. Since this latter ambiguity *always exists* (even in an ordinary nonstringy field theory!) it follows that off-shell string theory is, in this respect, no worse off than taking any other field theory off-shell.

From the worldsheet perspective, this ambiguity is also nothing special; it is just the usual scheme dependence found in RG theory. Again, this is something we are used to from a QFT perspective, so it shouldn't be taken as a special problem associated with string theory. The important relationship between renormalization of the worldsheet and propagation of strings in target space was explained in 4. This connects the off-shell formalism to the important observation by Susskind [14] that the UV cutoff in string theory acts as an IR cutoff in target space.

The other main issue, arising at tree-level, was the appropriate way to deal with the non-compact SL(2,$\mathbb{C}$) conformal Killing group on the sphere. We showed, from numerous perspectives, (including the S-matrix, the equations of motion, and c-theorems) that Tseytlin's $\partial/\partial \log \epsilon$ prescriptions[98] are a natural and acceptable way to deal with this CKG factor.

In our opinion, the most beautiful argument that the sphere prescription gives the correct S-matrix comes from our discussion of gauge-orbits in 5.1, but we also provided explicit discussions of what happens if you fix 3 points or 2 points (at generic momenta), in sections 5.2 and 5.3 respectively. We also showed how to extract the correct $i\varepsilon$ pole prescription in section 5.4.

Our arguments in section 6 that the correct equations of motion are obtained are a little less general, since we needed to assume a *renormalizability condition* on the allowed dimensions

---

[98]The word "prescriptions" is plural, because there are two of them, with **T2** having a broader range of applicability than **T1**.

of perturbed operators (section 6.2). Although Tseytlin usually works at leading order in $\alpha'$, we were able to re-phrase our results as an (order $n$-dependent) finite range of acceptable operator dimensions $\Delta$, where in particular both prescriptions have an upper bound on the degree of irrelevance that can be considered. At specified $n$, this is a little stronger than all orders in perturbation in $\alpha'$. Hence, our results for the tree-level EOM are proven to be correct to all orders in $g_s$ and $\alpha'$ (and in some special cases nonperturbatively in $\alpha'$).

It may be that a better understanding of c-theorems *on the sphere* would enable us to drop some of these remaining restrictions. This could enable a proof that the off-shell action works nonperturbatively, or in the presence of massive string excitations. But at least one new idea seems required to make this work. At the present moment of time we have only a perturbative C-function defined on the sphere (7).

In part II of this work [7], we will explain the underlying conceptual structure of the S&U black hole entropy argument. There we show explicitly how the effective action $I_0$ and the entropy $S = A/4G_N$ may be calculated from the sphere diagrams. We also discuss the behavior of the S&U entropy under RG flow. Although the conical manifold smooths out under RG flow, moving towards an on-shell configuration, the entropy doesn't change.

We will also compare these off-shell results with the much more popular *orbifold method* for calculating entropy from the on-shell $\mathbb{C}/Z_N$ background [1, 83, 113, 114]. By considering processes involving twisted string states, we will conclude that the orbifold method is physically incorrect—unless one allows tachyons to condense on the orbifold, in which case it appears (though the off-shell string field theory calculations are difficult and we did not attempt them ourselves) that one probably ends up back in the flowing cone scenario. However, there may be some important insights into the ER=EPR hypothesis that can be obtained from the fact that this condensate at a codimension-2 surface is apparently equivalent to ordinary flat space.

## Acknowledgments

We are grateful for conversations with Edward Witten, Arkady Tseytlin, Gabriel Wong, William Donnelly, Ronak Soni, Juan Maldacena, Donald Marolf, Raghu Mahajan, Lorenz Eberhardt, Eva Silverstein, Daniel Jafferis, Xi Yin, Lenny Susskind, Alexander Frenkel, Vasudev Shyam, Ayshalynne Abdel-Aziz, Zihan Yan, Houwen Wu, David Tong, and David Skinner. A.A. would like in particular to thank Prahar Mitra for extensive, very long and insightful discussions. A.W. would also like to thank Joe Polchinski for pointing him in the direction of Tseytlin's work, several years before he had the capacity to actually understand it.

**Funding information**   This work was supported in part by AFOSR grant FA9550-19-1-0260 "Tensor Networks and Holographic Spacetime", STFC grant ST/P000681/1 "Particles, Fields and Extended Objects", and an Isaac Newton Trust Early Career grant.

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
