# Peer review of "Off-Shell Strings I: S-matrix and Action"

_SciPost Physics, doi:SciPost Phys. 17, 005 (2024)_

## Round 1 · Referee Report · Anonymous (Referee 1) · 2023-9-22

Strengths

See acompanying file

Weaknesses

See accompanying file

Report

See accompanying file

Requested changes

See accompanying file

Attachment

  • validity: top
  • significance: top
  • originality: top
  • clarity: good
  • formatting: excellent
  • grammar: excellent

Author:  Amr Ahmadain  on 2024-05-17  [id 4490]

(in reply to Report 1 on 2023-09-22)
Category:
remark
answer to question
reply to objection

Dear Referee,

We have addressed all of your comments, questions and remarks to the best of our ability. We have made significant changes to section III.C and section VII especially VII.A, VII.B and VII.C. This is in addition to several other minor changes to the whole text.

The attached PDF file contains a detailed 6-page exposition of the changes made to the text and responses to all of your questions and remarks. Our replies are the blue-colored text in the PDF file.

If you still have any further questions or remarks, we'll be happy to address them.

The Authors

Attachment:

Off_shell_Strings_I_SciPost_Response_to_Anonymous_Referee.pdf

Anonymous on 2024-05-22  [id 4505]

(in reply to Amr Ahmadain on 2024-05-17 [id 4490])

I am satisfied with the corrections that have been made and do not think a further round of refereeing would be useful.

---

## Round 1 · Referee Report · Matthew Headrick (Referee 2) · 2023-12-14

Strengths

See report.

Weaknesses

See report.

Report

This paper takes up the long-dormant mantle of Tseytlin’s nonlinear sigma model approach to string theory. Many technical advances are made, and the whole theory is put on a somewhat more secure foundation. In addition, the theory is explained in a more transparent way than in Tseytlin’s many papers on the subject, which unfortunately suffered from leaps in logic, hidden assumptions, etc. Applications of the theory, in particular to black hole entropy, come in a second paper, which I am not reviewing here.

This paper is long and highly technical, and addresses many subtle and confusing issues. While I think I understand the gist, and did not find any suspicious or outright false claims, I cannot claim to have checked each derivation carefully. Nonetheless, based on what I do understand, I believe the paper easily clears the bar for publication in SciPost. The results are of great importance for our understanding of string theory, and, with some exceptions detailed below, the presentation is generally clear.

Requested changes

Before publication, I would like the authors to address the presentational issues listed below. Some of these are minor or cosmetic, while others are more substantive. In the cases where I suggest a fix, based on my understanding, the authors don’t have to follow my suggestion; but in all cases they need to address the issue. From p. 18 onward, where my list ends, the authors may want to follow the spirit of the suggestions and try to identify and clean up any further presentational infelicities.

p. 2 R column, a few lines below (T1), “super(string) theory”, why is “string” in parentheses?

p. 2 R column, near the bottom, first bullet: What does “the limit where $\log\epsilon^{-1}$ is small” mean? $\epsilon$ is dimensionful, so I don’t think you mean the limit $\epsilon\to1$. I think you just mean “at finite $\epsilon$”, i.e. not taking the limit of the next bullet.

p. 3 R column, near bottom: “The sigma model approach is most successful only when the characteristic length of the background spacetime is much less than the string scale”. Don’t you mean “greater”?

p. 3 4 R column, just below (3): “Unfortunately, this method does not give the correct entropy unless perhaps (following Dabholkar [82]) we allow tachyons to condense on the orbifold.” Perhaps the authors did not intend it this way, but to my reading this is a weirdly derogatory and dismissive throw-away comment, toward what many of us believe is an interesting and well-grounded line of research. Why “perhaps”? Why would we not allow tachyons to condense? Obviously this is not the place for a full discussion of these issues, which presumably comes in paper II. I would suggest just deleting this sentence (and maybe citing Dabholkar in the previous one).

p. 5 L column, top of page: “For products over $n$…” This really confused me. I think you don’t mean products “over $n$”, you mean products over the vertex operators at fixed $n$. The notation strongly suggests a product over $n$, making equations like (22), (30), etc needlessly hard to understand. I realize you don’t want to include yet another index, but some change of notation would be helpful. Maybe put the $n$ over (rather than under) the $\Pi$, since it is a product “up to $n$”?

p. 7 L column, bottom of page: “i.e. is proportional to some $E_A$” Shouldn’t that be $E_a$?

p. 9 R column: Eq (31) is impossible to understand. What does the colon mean? What is on the LHS of the equation? Please rewrite using standard notation.

p. 12 R column: I didn’t understand in what sense the S-matrix emerges in the limit that the effective action becomes non-local. Usual QFTs have a local action and an S-matrix. Related to this, my understanding was that the worldsheet cutoff $\epsilon$ is related to the size of the string: in the limit the cutoff is small, the string gets large and the effective action becomes non-local in the target space. However, here it seems to be related instead to the distance over which the string can propagate. What is the relation between these things?

p. 13 R column: Eq (40) is missing a minus sign in the exponent.

p. 14 L column: The measure factor in parentheses is confusing, with the $n$ subscript. Maybe just write $d^{2n}z$?

p. 15 caption to fig 5(i): “the hyperbolic volume of the regulated gauge orbit is noncompact” I think you mean “is infinite”.

p. 17 L column: The notation $ij\ldots z$ is confusing, given the other role of $z$ here. I would recommend instead $i_1\ldots i_n$ (particularly since the number $n$ of them is fixed).

p. 17 R column: On the LHS of (60), I believe that $I_0^{eff}$ should be $I_{(\chi)}$.

p. 17: Eq (61) follows directly from (57) and (58). I didn’t understand what was supposed to be gained by the detour through (59) and (60).

  • validity: top
  • significance: top
  • originality: top
  • clarity: high
  • formatting: perfect
  • grammar: perfect

Author:  Amr Ahmadain  on 2024-05-20  [id 4496]

(in reply to Report 2 by Matthew Headrick on 2023-12-14)
Category:
remark
answer to question
reply to objection

Dear Matthew,

We have addressed all of your comments, questions and remarks to the best of our ability.

The attached PDF file contains a detailed 3-page exposition of the changes made to the text and responses to all of your questions and remarks. Our replies are the blue-colored text in the PDF file.

If you still have any further questions or remarks, we'll be happy to address them.

The Authors

Attachment:

Off_shell_Strings_I_SciPost_Response_to_Matt_Headrick.pdf

---

## Round 2 · Author Response

Dear Editor-in-charge,

Here is the revised version of your manuscript which contains all the changes described in the PDF files attached as individual replies to the referees.

We kindly ask you to recommend the paper for publication to the editorial college.

Best regards,
The Authors

---

## Round 2 · List of Changes

Please see the PDF files for detailed exposition of the changes made to the text. The PDF should be publicly visible.

---

## Editorial Decision

published